# Predictors of plasma leakage among dengue patients in Thailand: A plasma-leak score analysis

Sutopa Talukdar[1], Vipa Thanachartwet[1]*, Varunee Desakorn[1], Supat Chamnanchanunt[1], Duangjai Sahassananda[2], Mukda Vangveeravong[3], Siripen Kalayanarooj[3], Anan Wattanathum[4]

1 Faculty of Tropical Medicine, Department of Clinical Tropical Medicine, Mahidol University, Bangkok, Thailand, 2 Faculty of Tropical Medicine, Information Technology Unit, Mahidol University, Bangkok, Thailand, 3 Department of Medical Services, Queen Sirikit National Institute of Child Health, Ministry of Public Health, Bangkok, Thailand, 4 Department of Medicine, Pulmonary and Critical Care Division, Phramongkutklao Hospital, Bangkok, Thailand

* vipa.tha@mahidol.edu

**Data Availability Statement:** All relevant data are within the paper.

## Abstract

Delayed plasma leakage recognition could lead to improper fluid administration resulting in dengue shock syndrome, subsequently, multi-organ failure, and death. This prospective observational study was conducted in Bangkok, Thailand, between March 2018 and February 2020 to determine predictors of plasma leakage and develop a plasma leakage predictive score among dengue patients aged $\geq$15 years. Of 667 confirmed dengue patients, 318 (47.7%) developed plasma leakage, and 349 (52.3%) had no plasma leakage. Multivariate analysis showed three independent factors associated with plasma leakage, including body mass index $\geq$25.0 kg/m$^2$ (odds ratio [OR] = 1.784; 95% confidence interval [CI] = 1.040–3.057; P = 0.035), platelet count <100,000/mm$^3$ on fever days 3 to 4 (OR = 2.151; 95% CI = 1.269–3.647; P = 0.004), and aspartate aminotransferase or alanine aminotransferase $\geq$100 U/l on fever days 3 to 4 (OR = 2.189; 95% CI = 1.231–3.891; P = 0.008). Because these three parameters had evidence of equality, each independent factor was weighted to give a score of 1 with a total plasma-leak score of 3. Higher scores were associated with increased plasma leakage occurrence, with ORs of 2.017 (95% CI = 1.052–3.869; P = 0.035) for score 1, 6.158 (95% CI = 2.914–13.015; P <0.001) for score 2, and 6.300 (95% CI = 2.419–16.407; P <0.001) for score 3. The area under the receiver operating characteristics curves for predicting plasma leakage was good (0.677 [95% CI = 0.616–0.739]). Patients with a plasma-leak score $\geq$1 had high sensitivity (88.8%), and those with a plasma-leak score of 3 had high specificity (93.4%) for plasma leakage occurrence. This simple and easily accessible clinical score might help physicians provide early and timely appropriate clinical dengue management in endemic areas.

**Funding:** This study was supported by the Faculty of Tropical Medicine, Mahidol University in Bangkok, Thailand. The funding organizations had no role in the study design, data collection, data analysis, decision to publish or writing of the manuscript.

**Competing interests:** The authors declare that they have no competing interests.

## Introduction

Dengue caused by the dengue virus (DENV) and spread by *Aedes aegypti* mosquito is currently a global burden, and dengue cases have increased by 8 fold over the last 10 years, with 70% of the global burden from Asia [1]. The reports from the Bureau of Epidemiology of Thailand from 2001–2016 showed an annual dengue incidence of 62–238 cases per 100,000 population with indefinable trends. Approximately 60–70% of the patients were adults aged ≥15 years [2].

The case fatality rate decreased from 0.18 in 2001 to 0.10 in 2016, but the mortality rate was higher in adults aged >40 years, ranging from 0.19–0.30 [2]. A previous study showed dengue shock syndrome (DSS) as the most common cause of death in adults and children with dengue accounted for 73%, followed by severe organ involvement (69%) and severe bleeding (30%) [3]. Delayed plasma leakage recognition could lead to inappropriate fluid management, resulting in DSS and subsequent multi-organ failure and death [4–6]. In addition, a previous Brazilian study reported plasma leakage and organ failure as the main indications for dengue patients' hospitalization, and there was an association between plasma leakage and dengue mortality [7].

The World Health Organization (WHO) has proposed timely and appropriate clinical management, which involves dengue diagnosis and intravenous rehydration as the strategy to reduce dengue mortality to almost zero [4]. Several studies have attempted to investigate plasma leakage predictors among dengue patients aged ≥15 years. However, studies reported on different parameters, including demographic characteristics of older age [8–11], gender [8,11], ethnicity [11], diabetes mellitus [9,10–12], hypertension [11], delayed hospitalization [9], secondary infection [9], clinical parameters of bleeding [8], abdominal pain [8,10], lethargy [8,9], or cough [8], and laboratory findings of hematocrit (HCT) rising [12–14], thrombocytopenia [13,15], abnormal coagulation profile [14], raised liver enzymes [12,13], low serum albumin (ALB) level [13,15], or thickening of the gall bladder wall [9]. Recent studies have added several new parameters, including procalcitonin [16], lactate [16,17], chymase [18], and cytokines [19], as plasma leakage predictors among dengue patients aged ≥15 years.

Some of these laboratory parameters may not be accessible in remote and resource-limited settings, where patients at risk for plasma leakage need to be identified, using simple clinical assessment methods and easily accessible laboratory investigations to improve healthcare utilization efficiency and save patients from unnecessary expenditure, loss of productivity, morbidity, and mortality associated with dengue. Furthermore, no study developed a simple clinical score to determine plasma leakage, which might help physicians provide timely and appropriate clinical management for dengue in endemic areas.

Thus, a prospective observational study was conducted at the Hospital for Tropical Diseases in Bangkok, Thailand, between March 2018 and February 2020 to determine predictors of plasma leakage and develop a predictive score for plasma leakage among dengue patients aged ≥15 years.

## Materials and methods

### Ethical considerations

The study was approved by the Ethics Committee of the Faculty of Tropical Medicine, Mahidol University in Bangkok, Thailand (TMEC 17–084). This study followed the Strengthening the Reporting of Observational Studies in Epidemiology (STROBE) statement [20] and the Standards for the Reporting of Diagnostic (STARD) accuracy [21]. All potential participants who visited the outpatient department (OPD) or inpatient department (IPD) for the management of dengue were invited to participate in the study. Before participation, voluntary written

informed consent was obtained from all patients or the patient's guardians if they were 15–18 years old.

## Study design and population

This was a prospective observational study conducted at the Hospital for Tropical Diseases, Faculty of Tropical Medicine, Mahidol University in Bangkok, Thailand, between March 2018 and February 2020. Patients who visited OPD or IPD for dengue management and met the study criteria were included. The inclusion criteria were aged ≥15 years, being clinical dengue patients, defined as acute fever <7 days and having ≥2 of the symptoms, including headache, retro-orbital pain, myalgia, arthralgia or bone pain, rash, bleeding, leukopenia defined as white blood cell count (WBC) ≤5.0 ×$10^3$ cells/mm$^3$, thrombocytopenia defined as platelet (PLT) count ≤150 ×$10^3$/mm$^3$ or HCT rising ≥5% with positive DENV NS1 and/or anti-DENV IgM. The real-time reverse transcriptase-polymerase chain reaction (rRT-PCR) or micro-neutralization test was used to confirm dengue in all potential patients enrolled for this study. Patients who were not followed-up and those with poor blood specimen quality or errors in the pre-analytical process were excluded from the study.

The baseline characteristics data, comorbid conditions, symptoms, signs, and laboratory investigations were collected at the start of management. Then, follow-up information data were collected daily from patients treated at both OPD and IPD. At a 2-week follow-up appointment, blood samples were collected for complete blood counts and DENV infection confirmation. All data were recorded in a pre-designed case record form. All patients were managed by their attending physicians according to the standard guidelines for dengue management [5,6]. Tests were performed to obtain data on routine monitoring parameters, including the day of fever, clinical condition, vital signs, and complete blood count, during the follow up of the patients. Other tests, for data on additional laboratory parameters including liver enzymes, serum ALB, and chest radiography, were performed according to the attending physicians' instructions, as per the clinical condition of the patients. Urine output was recorded for patients treated at the IPD. Plasma leakage was summarized at discharge date and defined as a rise in HCT ≥20%, clinical fluid accumulation by detecting pleural effusion or ascites using physical examination or chest radiography, and/or hypoproteinemia determined by serum ALB ≤3.5 g/dl or decrease ≥0.5 g/dl below baseline [5].

## Real-time reverse-transcriptase polymerase chain reaction (rRT-PCR)

DENV RNA was detected from patient serum on the first day of presentation using a two-step PCR method, as described by Lanciotti *et al*. [22], and modified using the methods of Reynes *et al*. [23]. PureLink® Viral RNA/DNA Mini Kit (Invitrogen™, Grand Island, NY, USA) was used to detect Dengue RNA from acute serum samples according to the manufacturer's instructions.

## Micro-neutralization test

Serum samples collected at presentation to hospital and two weeks after the first presentation were assayed for serotype-specific DENV using the micro-neutralization test described by Vorndam *et al*. [24], with the slightly modified protocol of Putnak *et al*. [25]. The micro-neutralization test based on the principle of the plaque reduction neutralization test was used to measure all four serotype-specific anti-DENV neutralizing antibodies. Serum samples were tested in triplicate and serially diluted by 2-fold from 1:20 to 1:5120 in a 96-well microplate. Each microplate had controls, including media only (negative control), virus control, and sera of known specific DENV serotypes (positive controls). The virus control with viral foci counts

in the range of 50–60 foci per well and media only control with no foci were included. Compared to the virus control, sera of known DENV serotypes showing at least 50% viral replication inhibition was required (25–30 foci per well). The DENV neutralization titer was defined as the reciprocal of the serum dilution, demonstrating 50% inhibition of DENV replication compared with the DENV control. A positive serotype-specific anti-DENV test was defined as a 4-fold rise in neutralizing antibody titer in paired samples for DENV serotypes 1 to 4.

## Sample size calculation

A previous report showed a plasma leakage incidence of 36.4% dengue patients in Bangkok, Thailand [26]. Thus, a sample size of 640 patients was needed for this study, using an incidence rate of 36.4% for plasma leakage with an error allowance of 3.5% and estimating 15% loss to follow-up and inadequate leftover samples.

## Statistical analysis

All data were analyzed using SPSS software (version 18.0; SPSS Inc., Chicago, IL). Kolmogorov-Smirnov test was used to analyze for normal distribution of numerical variables. Variables with non-normal distribution were summarized as medians and interquartile ranges (IQRs) and compared using Mann-Whitney $U$ tests. Categorical variables were expressed as frequencies and percentages and analyzed using chi-square or Fisher's exact tests. A univariate logistic regression analysis was performed with each potential factor included as an independent variable and the presence or absence of plasma leakage as the dependent variable. Patients' characteristics, clinical and laboratory findings associated with plasma leakage development were categorized. The categorical parameters that were early, clinically significant, and not subjective were evaluated.

Any variable with a P ≤0.2 was considered potentially significant and was further analyzed in a stepwise multivariate logistic regression analysis using a backward selection method for determining significant independent factors. The optimal cut-off values of the factors for the prediction of plasma leakage were determined using the area under the receiver operating characteristics (AUROC) curves. Prognostic parameters were evaluated using 2×2 tables, and 95% confidence intervals (CIs) were calculated to determine sensitivity, specificity, positive predictive value (PPV), negative predictive value (NPV), positive likelihood ratio (LR+), and negative likelihood ratio (LR–). The optimal cut-off values were then combined in a single "score," as described by Gibot *et al.*, 2012 [27]. As per the score, one point was attributed to each independent factor's presence. The score was classified as 0 (absence of all independent factors), 1 (presence of one independent factor), 2 (presence of two independent factors), or 3 (presence of three independent factors). The score was then further tested by multivariate logistic regression analysis for predictive value in plasma leakage. All tests of significance were two-sided, with a P <0.05 indicating statistical significance.

## Results

### Study population

A total of 750 suspected dengue patients visited the hospital during the study period. Of 750 suspected dengue patients, 83 were excluded due to lack of leftover DENV infection samples for a confirmation test (29 patients), transfer to other hospital or loss to follow-up (18 patients), no comorbid conditions documentation (15 patients), negative DENV infection confirmatory results, using rRT-PCR and micro-neutralization test (12 patients), and lack of baseline laboratory parameters (9 patients). Therefore, 667 patients with confirmed DENV

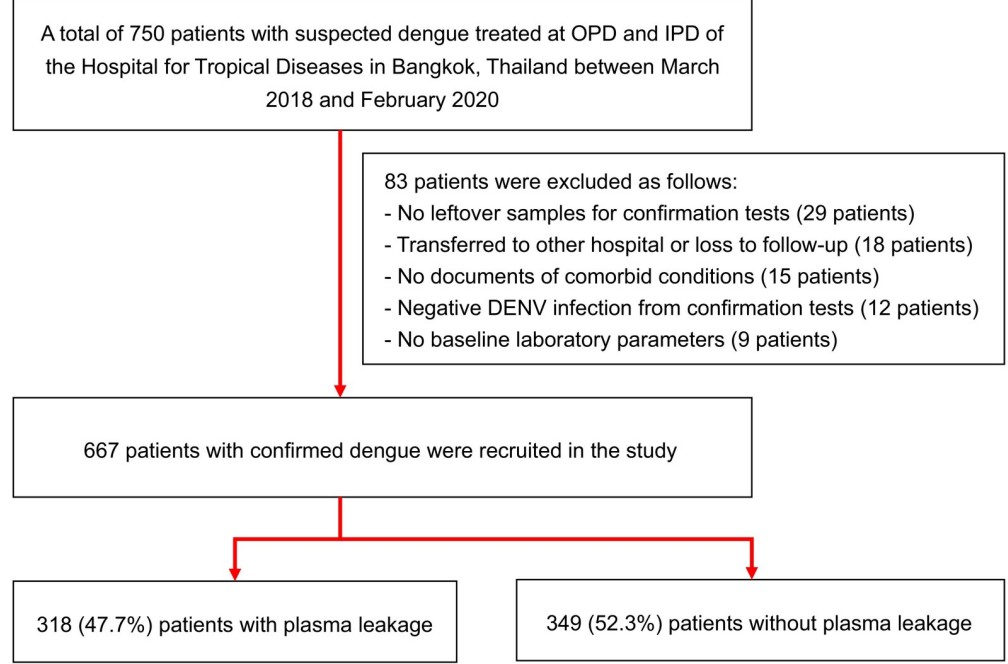

**Fig 1. Flow diagram showing the enrollment of study patients.** DENV, dengue virus; IPD, inpatient department; OPD, outpatient department.

infection were recruited in this study; 318 (47.7%) developed plasma leakage, whereas 349 (52.3%) had no plasma leakage (Fig 1). Only one (0.15%) patient expired due to DSS, severe bleeding, and multi-organ failure during this study; of 318 patients with plasma leakage, 14 (4.4%) patients developed DSS.

## Comparison of patients' characteristics, dengue serotypes, clinical and laboratory findings between dengue patients with and without plasma leakage

Most patients' characteristics were similar (Table 1), except for a significantly higher proportion of males among patients with plasma leakage than those without plasma leakage (P = 0.010). Similarly, patients with plasma leakage had significantly higher body mass index (BMI) (P = 0.009). Referral patients were found more among patients with plasma leakage than those without, which was significant (P = 0.006). However, dengue serotypes among patients with plasma leakage and those without were similar (P = 0.076).

A significantly higher proportion of patients with plasma leakage were admitted in comparison to those without plasma leakage (P <0.001). In addition, a significant delay in hospitalization (P <0.001) and longer hospitalization duration (P <0.001) were more frequently observed among plasma leakage patients than those without plasma leakage (Table 1). Clinical symptoms and signs of patients were similar in both groups on days 1 to 2 of fever onset (Table 2). However, symptoms and signs of vomiting (P = 0.012), bleeding (P = 0.030), and hepatomegaly (P = 0.006) were observed significantly more frequently among patients with plasma leakage than those without plasma leakage on days 3 to 4 of fever onset (Table 2). Abdominal pain (P = 0.024), vomiting (P = 0.035), bleeding (P <0.001), and hepatomegaly (P <0.001) on days 5 to 6 of fever onset was observed in a significantly higher proportion of plasma leakage patients than patients without plasma leakage (Table 2). However, on days 7 to

**Table 1. Patients' characteristics, dengue serotypes, and outcomes of plasma leakage status (n = 667).**

| Characteristic | Total (n = 667) | With plasma leakage (n = 318) | Without plasma leakage (n = 349) | P-value |
|---|---|---|---|---|
| Fever onset (days)[a] | 4.0 (3.0–5.0) | 4.0 (3.0–5.0) | 4.0 (3.0–5.0) | 0.058 |
| Age (years)[a] | 26 (20–37) | 27 (21–38) | 25 (20–37) | 0.091 |
| Male[b] | 348 (52.2) | 183 (57.5) | 165 (47.3) | 0.010 |
| BMI (kg/m$^2$)[a,c] | 23.0 (20.0–26.4) | 23.2 (20.5–27.0) | 22.7 (19.7–25.6) | 0.009 |
| Residence in Bangkok[b] | 470 (70.5) | 219 (68.9) | 251 (71.9) | 0.437 |
| Referral patients[b] | 308 (46.2) | 165 (51.9) | 143 (41.0) | 0.006 |
| Previous history of dengue[b] | 108 (16.2) | 53 (16.7) | 55 (15.8) | 0.832 |
| Comorbid condition[b] | 175 (26.2) | 90 (28.3) | 85 (24.4) | 0.285 |
| Hypertension[b] | 34 (5.1) | 20 (6.3) | 14 (4.0) | 0.246 |
| Thalassemia or G6PD deficiency[b] | 33 (4.9) | 20 (6.3) | 13 (3.7) | 0.178 |
| Hyperlipidemia[b] | 29 (4.3) | 14 (4.4) | 15 (4.3) | 1.000 |
| Diabetes mellitus[b] | 12 (1.8) | 6 (1.9) | 6 (1.7) | 1.000 |
| Peptic ulcer disease[b] | 11 (1.6) | 5 (1.6) | 6 (1.7) | 1.000 |
| HBV or HCV infection[b] | 7 (1.0) | 3 (0.9) | 4 (1.1) | 1.000 |
| HIV infection[b] | 5 (0.7) | 2 (0.6) | 3 (0.9) | 1.000 |
| Asthma[b] | 5 (0.7) | 2 (0.6) | 3 (0.9) | 1.000 |
| Confirmation test for dengue[b] | | | | |
| Serotype 1 | 247 (37.0) | 121 (38.1) | 126 (36.1) | 0.076 |
| Serotype 2 | 283 (42.4) | 145 (45.6) | 138 (39.5) | |
| Serotype 3 | 44 (6.6) | 16 (5.0) | 28 (8.0) | |
| Serotype 4 | 93 (13.9) | 36 (11.3) | 57 (16.3) | |
| Hospitalization[b] | 553 (82.9) | 304 (95.6) | 249 (71.3) | <0.001 |
| Delay in hospitalization[b] | 399 (59.8) | 225 (70.8) | 174 (49.9) | <0.001 |
| Duration of hospitalization[a,d] (days) | 3.5 (2.4–4.6) | 3.8 (2.8–4.8) | 2.9 (2.0–3.9) | <0.001 |

BMI, body mass index; G6PD, glucose-6-phosphate dehydrogenase; HBV, hepatitis B virus; HCV, hepatitis C virus; HIV, human immunodeficiency virus.

[a]Data presented as median (interquartile range).

[b]Data presented as number (percentage).

[c]Body mass index was measured in 640 patients, of which 304 patients had plasma leakage, and 336 patients had no plasma leakage.

[d]In a total of 553 hospitalized patients, plasma leakage was observed in 204 patients, and plasma leakage did not occur in 249.

8 of fever onset, retro-orbital pain was observed in a significantly lower percentage of plasma leakage patients (P = 0.032) while cough (P = 0.015), abdominal pain (P <0.001), bleeding (P <0.001), and hepatomegaly (P <0.001) were significantly higher (Table 2).

Regarding vital signs and cumulative fluid balance (Tables 3 and 4), body temperature and mean arterial pressure (MAP) were similar between the two groups on days 1 to 2 of fever onset. Body temperature on days 3 to 5 and day 7 of fever onset were significantly higher among patients with plasma leakage than those without plasma leakage with P <0.05. When stratified by combined day of fever, body temperature (Fig 2A) between days 1 to 8 of fever onset remained significant (P <0.05). MAP on day 4 and days 6 to 8 of fever onset and cumulative fluid balance on days 5 to 8 of fever onset were significantly higher among patients with plasma leakage than those without plasma leakage with P <0.05 (Tables 3 and 4).

Regarding laboratory findings (Tables 3 and 4), all laboratory findings including WBC, absolute lymphocyte count (ALC), HCT rise, PLT count, aspartate aminotransferase (AST), alanine aminotransferase (ALT), and serum ALB were similar on day 1 of fever onset. However, patients with plasma leakage had significantly higher WBC levels on days 6 to 8 of fever onset and higher ALC on days 5 to 6 of fever onset than those without plasma leakage

**Table 2. Symptoms and signs of confirmed dengue by day of fever onset and plasma leakage status (n = 667).**

| Characteristic | Days 1 to 2 of fever onset | | | | Days 3 to 4 of fever onset | | | | Days 5 to 6 of fever onset | | | | Days 7 to 8 of fever onset | | | |
|---|---|---|---|---|---|---|---|---|---|---|---|---|---|---|---|---|
| | Total (n = 123) | With PL (n = 59) | Without PL (n = 64) | P-value | Total (n = 453) | With PL (n = 231) | Without PL (n = 222) | P-value | Total (n = 624) | With PL (n = 312) | Without PL (n = 312) | P-value | Total (n = 558) | With PL (n = 283) | Without PL (n = 275) | P-value |
| Fever (%) | 123 (100) | 59 (100) | 64 (100) | N/A | 429 (94.7) | 219 (94.8) | 210 (94.6) | 1.000 | 434 (69.6) | 227 (72.8) | 207 (66.3) | 0.098 | 111 (19.9) | 51 (18.0) | 60 (21.8) | 0.309 |
| Myalgia (%) | 111 (90.2) | 51 (86.4) | 60 (93.8) | 0.289 | 392 (86.5) | 204 (88.3) | 188 (84.7) | 0.321 | 420 (67.3) | 209 (67.0) | 211 (67.6) | 0.932 | 165 (29.6) | 78 (27.6) | 87 (31.6) | 0.336 |
| Headache (%) | 107 (87.0) | 51 (86.4) | 56 (87.5) | 1.000 | 373 (82.3) | 190 (82.3) | 183 (82.4) | 1.000 | 351 (56.3) | 175 (56.1) | 176 (56.4) | 1.000 | 112 (20.1) | 51 (18.0) | 61 (22.2) | 0.262 |
| Chills (%) | 96 (78.0) | 45 (76.3) | 51 (79.7) | 0.811 | 293 (64.7) | 149 (64.5) | 144 (64.9) | 1.000 | 226 (36.2) | 112 (35.9) | 114 (36.5) | 0.934 | 20 (3.6) | 6 (2.1) | 14 (5.1) | 0.097 |
| Retro-orbital pain (%) | 66 (53.7) | 35 (59.3) | 31 (48.4) | 0.304 | 209 (46.1) | 111 (48.1) | 98 (44.1) | 0.459 | 202 (32.4) | 91 (29.2) | 111 (35.6) | 0.104 | 44 (7.9) | 15 (5.3) | 29 (10.5) | 0.032 |
| Nausea (%) | 64 (52.0) | 33 (55.9) | 31 (48.4) | 0.515 | 267 (58.9) | 139 (60.2) | 128 (57.7) | 0.654 | 257 (41.2) | 131 (42.0) | 126 (40.4) | 0.745 | 62 (11.1) | 31 (11.0) | 31 (11.3) | 1.000 |
| Cough (%) | 29 (23.6) | 13 (22.0) | 16 (25.0) | 0.861 | 147 (32.5) | 77 (33.3) | 70 (31.5) | 0.757 | 217 (34.8) | 119 (38.1) | 98 (31.4) | 0.093 | 140 (25.1) | 84 (29.7) | 56 (20.4) | 0.015 |
| Diarrhea (%) | 26 (21.1) | 17 (28.8) | 9 (14.1) | 0.075 | 140 (30.9) | 78 (33.8) | 62 (27.9) | 0.214 | 138 (22.1) | 76 (24.4) | 62 (19.9) | 0.210 | 54 (9.7) | 30 (10.6) | 24 (8.7) | 0.545 |
| Abdominal pain (%) | 23 (18.7) | 14 (23.7) | 9 (14.1) | 0.253 | 147 (32.5) | 82 (35.5) | 65 (29.3) | 0.189 | 222 (35.6) | 125 (40.1) | 97 (31.1) | 0.024 | 135 (24.2) | 87 (30.7) | 48 (17.5) | <0.001 |
| Vomiting (%) | 22 (17.9) | 13 (22.0) | 9 (14.1) | 0.359 | 119 (26.3) | 73 (31.6) | 46 (20.7) | 0.012 | 124 (19.9) | 73 (23.4) | 51 (16.3) | 0.035 | 22 (3.9) | 13 (4.6) | 9 (3.3) | 0.559 |
| SOB (%) | 18 (14.6) | 9 (15.3) | 9 (14.1) | 1.000 | 75 (16.6) | 37 (16.0) | 38 (17.1) | 0.851 | 83 (13.3) | 43 (13.8) | 40 (12.8) | 0.814 | 36 (6.5) | 14 (4.9) | 22 (8.0) | 0.195 |
| Sore throat (%) | 18 (14.6) | 10 (16.9) | 8 (12.5) | 0.658 | 88 (19.4) | 41 (17.7) | 47 (21.2) | 0.423 | 119 (19.1) | 51 (16.3) | 68 (21.8) | 0.103 | 53 (9.5) | 25 (8.8) | 28 (10.2) | 0.690 |
| Sneezing (%) | 17 (13.8) | 9 (15.3) | 8 (12.5) | 0.857 | 55 (12.1) | 26 (11.3) | 29 (13.1) | 0.656 | 53 (8.5) | 22 (7.1) | 31 (9.9) | 0.251 | 19 (3.4) | 13 (4.6) | 6 (2.2) | 0.181 |
| Bleeding (%) | 10 (8.1) | 6 (10.2) | 4 (6.2) | 0.518[a] | 98 (21.6) | 60 (26.0) | 38 (17.1) | 0.030 | 216 (34.6) | 134 (42.9) | 82 (26.3) | <0.001 | 210 (37.6) | 131 (46.3) | 79 (28.7) | <0.001 |
| Hepatomegaly (%) | 2 (1.6) | 1 (1.7) | 1 (1.6) | 1.000[a] | 19 (4.2) | 16 (6.9) | 3 (1.4) | 0.006 | 63 (10.1) | 49 (15.7) | 14 (4.5) | <0.001 | 51 (9.1) | 43 (15.2) | 8 (2.9) | <0.001 |
| Rash (%) | 2 (1.6) | 2 (3.4) | 0 (0) | 0.228[a] | 43 (9.5) | 26 (11.3) | 17 (7.7) | 0.252 | 139 (22.3) | 74 (23.7) | 65 (20.8) | 0.441 | 179 (32.1) | 98 (34.6) | 81 (29.5) | 0.223 |

N/A, not applicable; PL, plasma leakage; SOB, shortness of breath.

[a] Data were analyzed using Fisher's exact tests.

**Table 3. Vital signs, cumulative fluid balance, and laboratory findings of confirmed dengue by days 1 to 4 of fever onset and plasma leakage status (n = 667).**

| Characteristic | Day 1 of fever onset | | | | Day 2 of fever onset | | | | Day 3 of fever onset | | | | Day 4 of fever onset | | | |
|---|---|---|---|---|---|---|---|---|---|---|---|---|---|---|---|---|
| | Total | With PL | Without PL | P-value | Total | With PL | Without PL | P-value | Total | With PL | Without PL | P-value | Total | With PL | Without PL | P-value |
| Temp[a] (°C) | n = 35 | n = 17 | n = 18 | 0.134 | n = 99 | n = 50 | n = 49 | 0.066 | n = 256 | n = 131 | n = 125 | <0.001 | n = 419 | n = 220 | n = 199 | <0.001 |
| | 39.0 (38.5–39.4) | 39.0 (38.7–39.6) | 38.7 (38.1–39.1) | | 38.9 (38.3–39.4) | 39.0 (38.8–39.6) | 38.8 (38.2–39.3) | | 38.5 (38.0–39.2) | 38.7 (38.2–39.3) | 38.3 (37.8–39.0) | | 38.1 (37.5–38.8) | 38.2 (37.8–39.0) | 38.0 (37.8–38.6) | |
| MAP[a] (mmHg) | n = 35 | n = 17 | n = 18 | 0.568 | n = 98 | n = 50 | n = 48 | 0.062 | n = 256 | n = 131 | n = 125 | 0.061 | n = 419 | n = 220 | n = 199 | 0.036 |
| | 83 (79–96) | 83 (78–97) | 84 (80–92) | | 84 (77–93) | 86 (79–95) | 80 (76–92) | | 84 (76–94) | 85 (78–96) | 83 (75–93) | | 84 (76–92) | 86 (77–92) | 82 (75–91) | |
| Cumulative fluid[a] (ml/day) | n = 4 | n = 2 | n = 2 | N/A | n = 4 | n = 2 | n = 2 | N/A | n = 123 | n = 78 | n = 45 | 0.402 | n = 290 | n = 184 | n = 106 | 0.199 |
| | 480 (119–765) | 160 (78, 241) | 750 (720, 780) | | 610 (396–1551) | 610 (536, 683) | 1095 (350, 1840) | | 500 (63–880) | 540 (-42–922) | 371 (130–708) | | 500 (-55–1105) | 542 (-51–1238) | 407 (-73–966) | |
| WBC[a] (×10³ cells/mm³) | n = 27 | n = 10 | n = 17 | 0.414 | n = 93 | n = 48 | n = 45 | 0.252 | n = 256 | n = 129 | n = 127 | 0.987 | n = 416 | n = 218 | n = 198 | 0.375 |
| | 5.10 (4.10–7.00) | 4.60 (4.00–6.48) | 5.90 (4.00–7.70) | | 3.90 (2.90–5.50) | 3.75 (2.65–5.10) | 4.10 (3.35–5.65) | | 3.10 (2.48–4.20) | 3.10 (2.55–4.10) | 3.20 (2.40–4.20) | | 2.80 (2.30–3.80) | 2.90 (2.30–3.82) | 2.80 (2.20–3.72) | |
| ALC[a] (cells/mm³) | n = 19 | n = 7 | n = 12 | 0.592 | n = 65 | n = 34 | n = 31 | 0.654 | n = 196 | n = 106 | n = 90 | 0.353 | n = 382 | n = 198 | n = 184 | 0.304 |
| | 102 (41–141) | 90 (45–108) | 122 (0–176) | | 100 (32–202) | 99 (0–207) | 100 (44–207) | | 139 (76–224) | 137 (76–212) | 142 (76–236) | | 185 (92–331) | 185 (93–390) | 180 (89–306) | |
| HCT rise[a] (%) | n = 27 | n = 10 | n = 17 | 0.711 | n = 93 | n = 48 | n = 45 | 0.911 | n = 257 | n = 130 | n = 127 | 0.040 | n = 416 | n = 218 | n = 198 | <0.001 |
| | 0.25 (0–7.63) | 0.29 (0–9.74) | 0 (0–6.42) | | 2.88 (0–7.32) | 1.58 (0–7.59) | 3.31 (0–7.13) | | 5.22 (1.03–11.58) | 6.36 (1.37–13.36) | 4.50 (0.74–9.58) | | 6.73 (2.76–12.20) | 8.43 (4.70–15.01) | 5.11 (1.13–9.36) | |
| PLT count[a] (×10³/mm³) | n = 27 | n = 10 | n = 17 | 0.570 | n = 93 | n = 48 | n = 45 | 0.209 | n = 257 | n = 130 | n = 127 | <0.001 | n = 416 | n = 218 | n = 198 | <0.001 |
| | 193 (154–215) | 178 (148–215) | 198 (158–214) | | 152 (116–181) | 150 (110–172) | 161 (126–207) | | 116 (75–146) | 101 (64–142) | 125 (90–156) | | 85 (50–124) | 73 (39–101) | 106 (64–138) | |
| AST[a] (U/l) | n = 27 | n = 10 | n = 17 | 0.219 | n = 93 | n = 48 | n = 45 | 0.012 | n = 153 | n = 96 | n = 57 | <0.001 | n = 237 | n = 141 | n = 96 | <0.001 |
| | 21 (17–27) | 22 (18–29) | 19 (16–22) | | 37 (28–54) | 38 (30–61) | 28 (24–41) | | 50 (34–90) | 60 (38–118) | 36 (24–63) | | 80 (47–154) | 97 (57–178) | 58 (39–91) | |
| ALT[a] (U/l) | n = 27 | n = 10 | n = 17 | 0.125 | n = 93 | n = 48 | n = 45 | <0.001 | n = 154 | n = 97 | n = 57 | <0.001 | n = 237 | n = 141 | n = 96 | <0.001 |
| | 22 (15–24) | 22 (17–24) | 20 (14–21) | | 30 (23–40) | 32 (26–42) | 23 (17–27) | | 32 (21–56) | 37 (25–76) | 24 (16–37) | | 51 (28–103) | 54 (37–120) | 38 (24–62) | |
| ALB[a] (g/dl) | n = 24 | n = 10 | n = 14 | 0.349 | n = 66 | n = 48 | n = 18 | 0.008 | n = 85 | n = 58 | n = 27 | 0.656 | n = 128 | n = 92 | n = 36 | 0.001 |
| | 4.8 (4.5–5.0) | 4.8 (4.6–5.0) | 4.5 (4.3–5.2) | | 4.7 (4.5–4.9) | 4.7 (4.5–4.9) | 4.5 (4.3–4.6) | | 4.4 (4.2–4.7) | 4.4 (4.2–4.7) | 4.4 (4.3–4.9) | | 4.2 (3.9–4.5) | 4.0 (3.8–4.5) | 4.3 (4.2–4.6) | |

ALB, albumin; ALC, absolute lymphocyte count; ALT, alanine aminotransferase; AST, aspartate aminotransferase; HCT, hematocrit; MAP, mean arterial pressure; N/A, not applicable; PL, plasma leakage; PLT, platelets; Temp, temperature; WBC, white blood cell count.

[a]Data presented as median (interquartile range).

**Table 4. Vital signs, cumulative fluid balance, and laboratory findings confirmed dengue patients by day 5 to 8 of fever onset and plasma leakage status (n = 667).**

| Characteristic | Day 5 of fever onset | | | | Day 6 of fever onset | | | | Day 7 of fever onset | | | | Day 8 of fever onset | | | |
|---|---|---|---|---|---|---|---|---|---|---|---|---|---|---|---|---|
| | Total | With PL | Without PL | P-value | Total | With PL | Without PL | P-value | Total | With PL | Without PL | P-value | Total | With PL | Without PL | P-value |
| Temp[a] (°C) | n = 554 | n = 296 | n = 258 | 0.013 | n = 580 | n = 300 | n = 280 | 0.103 | n = 553 | n = 283 | n = 270 | 0.022 | n = 389 | n = 222 | n = 167 | 0.948 |
| | 37.7 (37.0–38.4) | 37.8 (37.2–38.5) | 37.5 (37.0–38.3) | | 37.2 (36.8–37.8) | 37.2 (36.8–37.8) | 37.0 (36.7–37.8) | | 36.8 (36.5–37.2) | 36.9 (36.6–37.8) | 36.8 (36.5–37.1) | | 36.6 (36.5–37.0) | 36.6 (36.5–37.0) | 36.6 (36.5–37.0) | |
| MAP[a] (mmHg) | n = 554 | n = 296 | n = 258 | 0.056 | n = 580 | n = 300 | n = 280 | 0.006 | n = 554 | n = 283 | n = 271 | 0.006 | n = 389 | n = 222 | n = 167 | 0.001 |
| | 81 (74–90) | 83 (75–91) | 80 (74–88) | | 79 (72–87) | 80 (73–87) | 77 (71–85) | | 78 (72–86) | 79 (73–88) | 76 (71–84) | | 78 (72–87) | 80 (73–89) | 77 (71–83) | |
| Cumulative fluid balance[a] (ml/day) | n = 456 | n = 276 | n = 180 | <0.001 | n = 521 | n = 296 | n = 225 | <0.001 | n = 547 | n = 305 | n = 242 | <0.001 | n = 548 | n = 305 | n = 243 | 0.003 |
| | 580 (-12–1349) | 670 (100–1645) | 296 (-162–966) | | 630 (-192–1535) | 915 (-36–1942) | 291 (-482–1005) | | 381 (-565–1470) | 600 (-362–1910) | 232 (-653–982) | | 250 (-711–1385) | 420 (-664–1682) | 100 (-780–936) | |
| WBC[a] (×10³ cells/mm³) | n = 548 | n = 293 | n = 255 | 0.231 | n = 566 | n = 294 | n = 272 | 0.014 | n = 544 | n = 280 | n = 264 | 0.002 | n = 383 | n = 221 | n = 162 | 0.017 |
| | 3.00 (2.30–4.30) | 3.10 (2.30–4.60) | 2.90 (2.30–3.90) | | 3.80 (2.70–5.32) | 4.20 (2.70–5.90) | 3.55 (2.70–4.90) | | 4.90 (3.50–6.60) | 5.10 (3.90–6.80) | 4.30 (3.22–6.28) | | 5.20 (4.20–6.70) | 5.30 (4.40–6.90) | 4.80 (3.88–6.40) | |
| ALC[a] (/mm³) | n = 517 | n = 284 | n = 233 | 0.016 | n = 556 | n = 292 | n = 264 | <0.001 | n = 540 | n = 279 | n = 261 | 0.239 | n = 380 | n = 219 | n = 161 | 0.591 |
| | 297 (160–692) | 333 (180–797) | 270 (144–599) | | 711 (336–1270) | 818 (401–1362) | 582 (266–1071) | | 912 (500–1516) | 943 (574–1496) | 876 (458–1602) | | 869 (518–1334) | 910 (525–1334) | 816 (510–1348) | |
| HCT rise[a] (%) | n = 549 | n = 294 | n = 255 | <0.001 | n = 567 | n = 295 | n = 272 | <0.001 | n = 545 | n = 281 | n = 264 | <0.001 | n = 384 | n = 222 | n = 162 | 0.070 |
| | 7.61 (3.03–13.38) | 10.59 (4.87–16.32) | 5.50 (2.16–9.00) | | 7.50 (3.01–12.27) | 9.84 (4.52–16.11) | 5.66 (1.70–9.09) | | 5.22 (1.39–10.15) | 6.84 (2.36–12.20) | 3.78 (0.90–8.00) | | 3.48 (0–7.69) | 3.75 (0–9.06) | 3.03 (0–6.94) | |
| PLT count[a] (×10³/mm³) | n = 548 | n = 293 | n = 255 | <0.001 | n = 580 | n = 300 | n = 280 | <0.001 | n = 544 | n = 280 | n = 264 | <0.001 | n = 383 | n = 221 | n = 162 | <0.001 |
| | 61 (31–97) | 46 (23–76) | 82 (47–119) | | 45 (25–74) | 34 (19–55) | 62 (36–94) | | 50 (29–78) | 39 (24–60) | 67 (39–73) | | 64 (40–90) | 57 (35–89) | 72 (48–94) | |
| AST[a] (U/l) | n = 269 | n = 167 | n = 102 | <0.001 | n = 238 | n = 152 | n = 86 | 0.002 | n = 179 | n = 124 | n = 55 | 0.003 | n = 131 | n = 95 | n = 36 | 0.004 |
| | 116 (58–212) | 135 (79–270) | 72 (42–132) | | 128 (76–232) | 154 (88–267) | 107 (64–180) | | 141 (80–253) | 152 (97–260) | 89 (55–199) | | 142 (59–267) | 165 (88–315) | 72 (40–211) | |
| ALT[a] (U/l) | n = 269 | n = 167 | n = 102 | <0.001 | n = 238 | n = 152 | n = 86 | 0.001 | n = 179 | n = 124 | n = 55 | 0.016 | n = 130 | n = 95 | n = 35 | 0.166 |
| | 62 (34–126) | 72 (45–127) | 42 (23–91) | | 77 (47–153) | 88 (50–176) | 62 (38–102) | | 99 (50–178) | 108 (56–181) | 70 (34–159) | | 117 (65–230) | 135 (70–228) | 98 (52–246) | |
| ALB[a] (g/dl) | n = 170 | n = 125 | n = 45 | <0.001 | n = 165 | n = 125 | n = 40 | <0.001 | n = 121 | n = 98 | n = 23 | <0.001 | n = 75 | n = 58 | n = 17 | <0.001 |
| | 4.0 (3.7–4.3) | 3.9 (3.5–4.2) | 4.3 (3.8–4.6) | | 3.8 (3.4–4.1) | 3.6 (3.3–4.0) | 4.2 (4.0–4.6) | | 3.7 (3.4–4.0) | 3.6 (3.3–3.9) | 4.2 (4.0–4.4) | | 3.8 (3.4–4.2) | 3.6 (3.4–4.0) | 4.3 (4.2–4.8) | |

ALB, albumin; ALC, absolute lymphocyte count; ALT, alanine aminotransferase; AST, aspartate aminotransferase; HCT, hematocrit; MAP, mean arterial pressure; PL, plasma leakage; PLT, platelets; Temp, temperature; WBC, white blood cell count.

[a]Data presented as median (interquartile range).

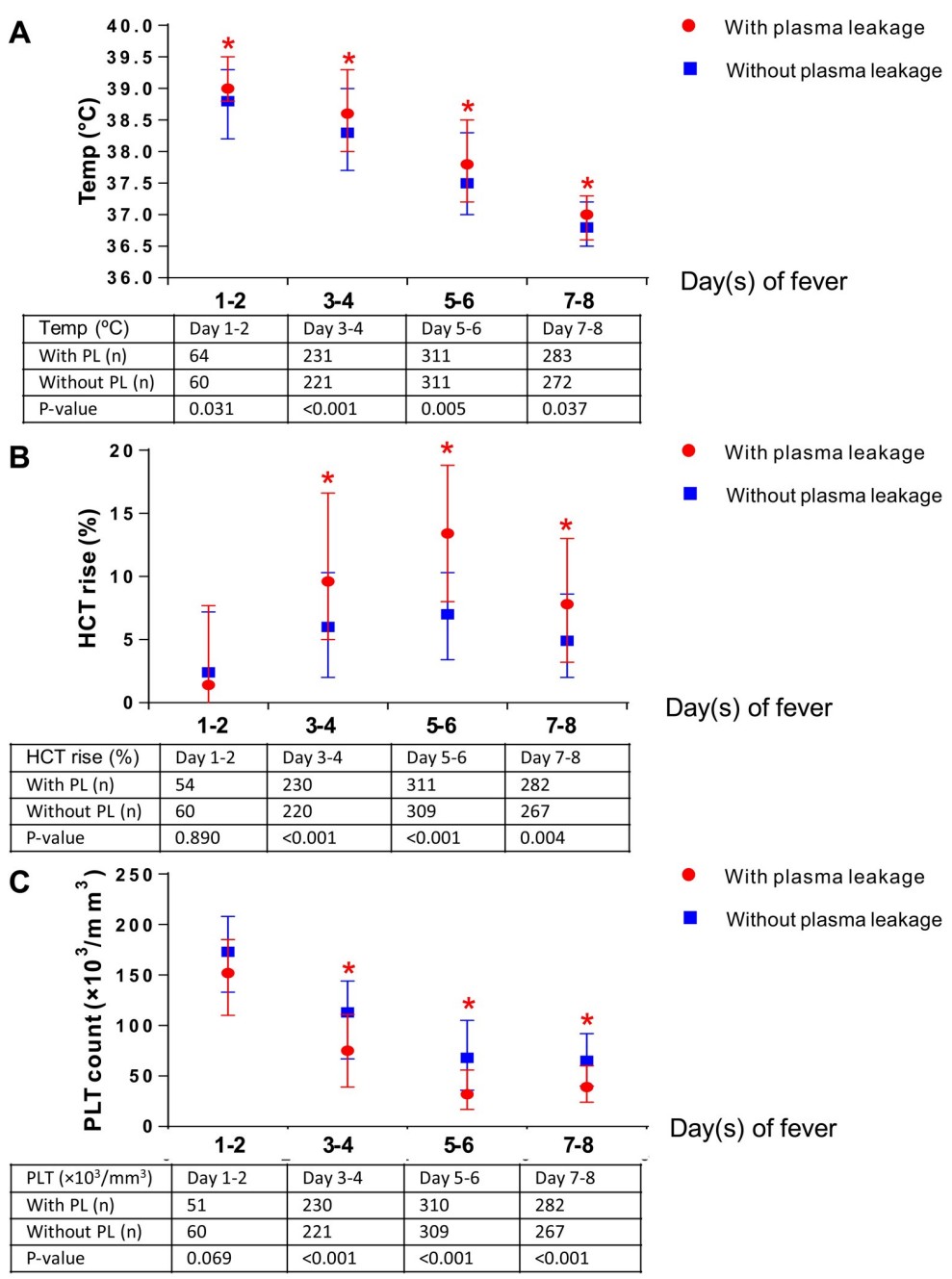

**Fig 2. Changes in body temperature, hematocrit rise, and platelet count among confirmed dengue patients by combined day of fever onset.** (a) The higest body temperature (˚C) by combined day of fever onset among patients with and without plasma leakage. (b). The higest hematocrit rise (%) by combined day of fever onset among patients with and without plasma leakage. (c). The lowest platelet count ($\times 10^3$/mm$^3$) by combined day of fever onset among patients with and without plasma leakage. HCT, hematocrit; PL, plasma leakage; PLT, platelets; Temp, temperature.

(P <0.05). HCT rise (Tables 3 and 4) between days 3 to 7 was significantly higher among patients with plasma leakage (P <0.05). When stratified by combined day of fever, HCT rise (Fig 2B) between days 3 to 8 of fever onset was also significantly higher among patients with plasma leakage (P <0.05). However, PLT count (Tables 3 and 4) between days 3 to 8 of fever

onset was significantly lower among patients with plasma leakage (P <0.001). When stratified by combined day of fever, PLT count (Fig 2C) between days 3 to 8 of fever was also significantly lower among plasma leakage patients (P <0.001). The levels of the liver enzymes, including AST and ALT (Tables 3 and 4) between days 2 to 8 of fever onset, when stratified by combined day of fever (Fig 3A and 3B) between days 1 to 8, were significantly higher among patients with plasma leakage than those without plasma leakage (P <0.05). Conversely, serum ALB levels (Tables 3 and 4) between days 4 to 8 of fever onset, when stratified by combined day of fever (Fig 3C) between days 3 to 8 were significantly lower among patients with plasma leakage than those without plasma leakage (P <0.05). However, serum ALB levels (Table 3) on day 2 of fever onset, when stratified by combined day of fever (Fig 3C) on days 1 to 2, were significantly higher among patients with plasma leakage than those without plasma leakage (P = 0.008 and P = 0.020, respectively).

## Univariate and multivariate analysis to predict the development of plasma leakage

Patients' characteristics, clinical and laboratory findings associated with development of plasma leakage were then categorized (Table 5) and evaluated using univariate logistic regression analysis. The following factors were found to be associated with the development of plasma leakage including patients' characteristics (male gender, BMI $\geq$25.0 kg/m$^2$, and delay in hospitalization); the symptoms and signs on fever days 3 to 4 (vomiting, bleeding, hepatomegaly, and body temperature >38.5˚C); the symptoms and signs on fever days 5 to 6 (abdominal pain); and the laboratory findings on days 3 to 4 of fever onset (HCT rise $\geq$10%, PLT count <100,000/mm$^3$, and AST or ALT $\geq$100 U/l) (Table 6).

The independent factors associated with plasma leakage identified by stepwise multiple logistic regression analysis were BMI $\geq$25.0 kg/m$^2$ (odds ratio [OR] = 1.784; 95% CI = 1.040–3.057; P = 0.035), PLT count <100,000/mm$^3$ on days 3 to 4 of fever onset (OR = 2.151; 95% CI = 1.269–3.647; P = 0.004), and AST or ALT $\geq$100 U/l on days 3 to 4 of fever onset (OR = 2.189; 95% CI = 1.231–3.891; P = 0.008) (Table 6).

## Plasma-leak score for predicting plasma leakage

Three factors, including BMI $\geq$25.0 kg/m$^2$, PLT count <100,000/mm$^3$, and AST or ALT $\geq$100 U/l were used to develop a score for predicting plasma leakage called a plasma-leak score. Because these three parameters had evidence of equality, each of the three independent factors were weighted to give a score of 1 with a total score of 3. A combined score was then evaluated using a logistic regression model to forecast the capacity for predicting the occurrence of plasma leakage. Higher scores were associated with increased occurrence of plasma leakage, with ORs of 2.017 (95% CI = 1.052–3.869; P = 0.035), 6.158 (95% CI = 2.914–13.015; P <0.001) and 6.300 (95% CI = 2.419–16.407; P <0.001) for a combined score of 1, 2, and 3, respectively (Table 7). The AUROC for predicting plasma leakage was 0.677 (95% CI = 0.616–0.739) (Fig 4).

## Prognostic values of plasma-leak score for identifying plasma leakage

The prognostic values of the plasma-leak score for identifying plasma leakage are summarized in Table 8. The sensitivity was 88.8% (95% CI = 83.2–93.0%) for a score $\geq$1. The specificities increased to 77.7% (95% CI = 69.2–84.8%) and 93.4% (95% CI = 87.4–97.1%) for a score of $\geq$2 and 3, respectively. Similarly, the PPVs increased to 77.5% (95% CI = 70.6–83.2%) and 77.8% (95% CI = 62.3–88.1%) for a score of $\geq$2 and 3, respectively.

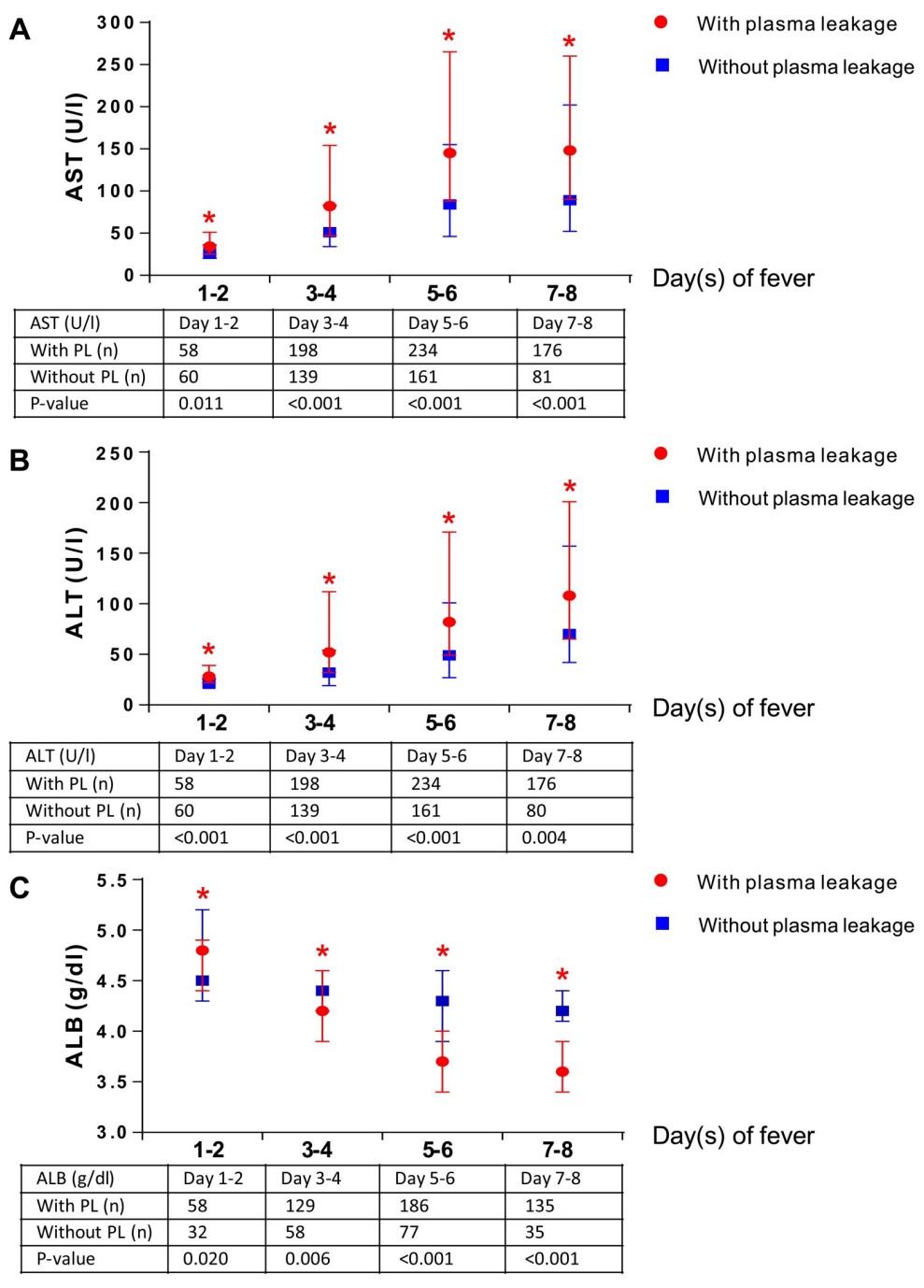

| AST (U/l) | Day 1-2 | Day 3-4 | Day 5-6 | Day 7-8 |
|---|---|---|---|---|
| With PL (n) | 58 | 198 | 234 | 176 |
| Without PL (n) | 60 | 139 | 161 | 81 |
| P-value | 0.011 | <0.001 | <0.001 | <0.001 |

| ALT (U/l) | Day 1-2 | Day 3-4 | Day 5-6 | Day 7-8 |
|---|---|---|---|---|
| With PL (n) | 58 | 198 | 234 | 176 |
| Without PL (n) | 60 | 139 | 161 | 80 |
| P-value | <0.001 | <0.001 | <0.001 | 0.004 |

| ALB (g/dl) | Day 1-2 | Day 3-4 | Day 5-6 | Day 7-8 |
|---|---|---|---|---|
| With PL (n) | 58 | 129 | 186 | 135 |
| Without PL (n) | 32 | 58 | 77 | 35 |
| P-value | 0.020 | 0.006 | <0.001 | <0.001 |

**Fig 3. Changes in aspartate aminotransferase, alanine aminotransferase, and serum albumin levels among confirmed dengue patients by combined day of fever onset.** (a) The highest level of AST (U/l) by combined day of fever onset among patients with and without plasma leakage. (b) The highest level of ALT (U/l) by combined day of fever onset among patients with and without plasma leakage. (c) The lowest level of serum ALB (g/dl) by combined day of fever onset among patients with and without plasma leakage. ALB, albumin; ALT, alanine aminotransferase; AST, aspartate aminotransferase; PL, plasma leakage.

**Table 5. Categorical data of patients' characteristics, clinical, and laboratory findings included in logistic regression analysis.**

| Characteristics | Total | With plasma leakage | Without plasma leakage | P-value |
|---|---|---|---|---|
| *Patients' characteristics* | | | | |
| Gender (%) | n = 667 | n = 318 | n = 349 | |
| Male | 348 (52.2) | 183 (57.5) | 165 (47.3) | 0.010 |
| Female | 319 (47.8) | 135 (42.5) | 184 (52.7) | |
| BMI (%) | n = 640 | n = 304 | n = 336 | |
| $\geq$25.0 kg/m$^2$ | 211 (33.0) | 114 (37.5) | 97 (28.9) | 0.025 |
| $<$25.0 kg/m$^2$ | 429 (67.0) | 190 (62.5) | 239 (71.1) | |
| Delay in hospitalization (%) | n = 667 | n = 318 | n = 349 | |
| Yes | 399 (59.8) | 225 (70.8) | 174 (49.9) | $<$0.001 |
| No | 268 (40.2) | 93 (29.2) | 175 (50.1) | |
| *Symptoms and signs on fever days 3 to 4* | | | | |
| Vomiting (%) | n = 453 | n = 231 | n = 222 | |
| Yes | 119 (26.3) | 73 (31.6) | 46 (20.7) | 0.012 |
| No | 334 (73.7) | 158 (68.4) | 176 (79.3) | |
| Bleeding (%) | n = 453 | n = 231 | n = 222 | |
| Yes | 98 (21.6) | 60 (26.0) | 38 (17.1) | 0.030 |
| No | 355 (78.4) | 171 (74.0) | 184 (82.9) | |
| Hepatomegaly (%) | n = 453 | n = 231 | n = 222 | |
| Yes | 19 (4.2) | 16 (6.9) | 3 (1.4) | 0.006 |
| No | 434 (95.8) | 215 (93.1) | 219 (98.6) | |
| Temp (%) | n = 452 | n = 231 | n = 221 | |
| $>$38.5˚C | 201 (44.5) | 117 (50.6) | 84 (38.0) | 0.009 |
| $\leq$38.5˚C | 251 (55.5) | 114 (49.4) | 137 (62.0) | |
| *Symptoms and signs on fever days 5 to 6* | | | | |
| Abdominal pain (%) | n = 624 | n = 312 | n = 312 | |
| Yes | 222 (35.6) | 125 (40.1) | 97 (31.1) | 0.024 |
| No | 402 (64.4) | 187 (59.9) | 215 (68.9) | |
| *Laboratory findings on fever days 3 to 4* | | | | |
| HCT rise (%) | n = 451 | n = 230 | n = 221 | |
| $\geq$10% | 181 (40.1) | 116 (50.4) | 65 (29.4) | $<$0.001 |
| $<$10% | 270 (59.9) | 114 (49.6) | 156 (70.6) | |
| PLT count (%) | n = 451 | n = 230 | n = 221 | |
| $<$100,000/mm$^3$ | 252 (55.9) | 163 (70.9) | 89 (40.3) | $<$0.001 |
| $\geq$100,000/mm$^3$ | 199 (44.1) | 67 (29.1) | 132 (59.7) | |
| AST or ALT (%) | n = 337 | n = 198 | n = 139 | |
| $\geq$100 U/l | 75 (22.3) | 60 (30.3) | 15 (10.8) | $<$0.001 |
| $<$100 U/l | 262 (77.7) | 138 (69.7) | 124 (89.2) | |

ALT, alanine aminotransferase; AST, aspartate aminotransferase; BMI, body mass index; HCT, hematocrit; PLT, platelets; Temp, temperature.

## Discussion

This prospective observational study was conducted at the Hospital for Tropical Diseases in Bangkok, Thailand, among dengue patients aged $\geq$15 years between March 2018 and February 2020 to determine predictors and a predictive score for plasma leakage. The median age of 26 years in the patients is similar to that of a previous report with a mean age of 30 years in adults

**Table 6. Univariate and multivariate logistic regression to determine independent risk factors associated with plasma leakage among dengue patients.**

| Characteristics | Univariate logistic regression analysis | | | Multivariate logistic regression analysis | | |
|---|---|---|---|---|---|---|
| | n | OR (95% CI) | P-value | n | OR (95% CI) | P-value |
| Gender | 667 | | | | | |
| Male | | 1.512 (1.113–2.053) | 0.008 | | | |
| Female | | Reference | | | | |
| BMI | 640 | | | 293 | | |
| $\geq 25.0$ kg/m$^2$ | | 1.478 (1.062–2.058) | 0.021 | | 1.784 (1.040–3.057) | 0.035 |
| $< 25.0$ kg/m$^2$ | | Reference | | | Reference | |
| Delay in hospitalization | 667 | | | | | |
| Yes | | 2.433 (1.767–3.351) | <0.001 | | | |
| No | | Reference | | | | |
| Vomiting | 453 | | | | | |
| Yes | | 1.768 (1.153–2.709) | 0.009 | | | |
| No | | Reference | | | | |
| Bleeding | 453 | | | | | |
| Yes | | 1.699 (1.076–2.682) | 0.023 | | | |
| No | | Reference | | | | |
| Hepatomegaly | 453 | | | 293 | | |
| Yes | | 5.433 (1.561–13.912) | 0.008 | | 4.042 (0.857–19.055) | 0.078 |
| No | | Reference | | | Reference | |
| Abdominal pain | 453 | | | | | |
| Yes | | 1.482 (1.065–2.060) | 0.019 | | | |
| No | | Reference | | | | |
| Temp | 452 | | | | | |
| >38.5˚C | | 1.674 (1.151–2.434) | 0.007 | | | |
| ≤38.5˚C | | Reference | | | | |
| HCT rise | 451 | | | | | |
| ≥10% | | 2.442 (1.657–3.600) | <0.001 | | | |
| <10% | | Reference | | | | |
| PLT count | 451 | | | 293 | | |
| <100,000/mm$^3$ | | 3.608 (2.440–5.337) | <0.001 | | 2.151 (1.269–3.647) | 0.004 |
| ≥100,000/mm$^3$ | | Reference | | | Reference | |
| AST or ALT | 337 | | | 293 | | |
| ≥100 U/l | | 3.202 (1.921–5.339) | <0.001 | | 2.189 (1.231–3.891) | 0.008 |
| <100 U/l | | Reference | | | Reference | |

ALT, alanine aminotransferase; AST, aspartate aminotransferase; BMI, body mass index; CI, confidence interval; HCT, hematocrit; OR, odds ratio; PLT, platelets; Temp, temperature.

**Table 7. Multivariate logistic regression of a combined plasma-leak score for predicting plasma leakage.**

| Score | n | Odds ratio (95% CI) | P-value |
|---|---|---|---|
| 0 | 56 | 1.00 (Reference) | N/A |
| 1 | 123 | 2.017 (1.052–3.869) | 0.035 |
| 2 | 84 | 6.158 (2.914–13.015) | <0.001 |
| 3 | 36 | 6.300 (2.419–16.407) | <0.001 |

CI, confidence interval; N/A, not applicable.

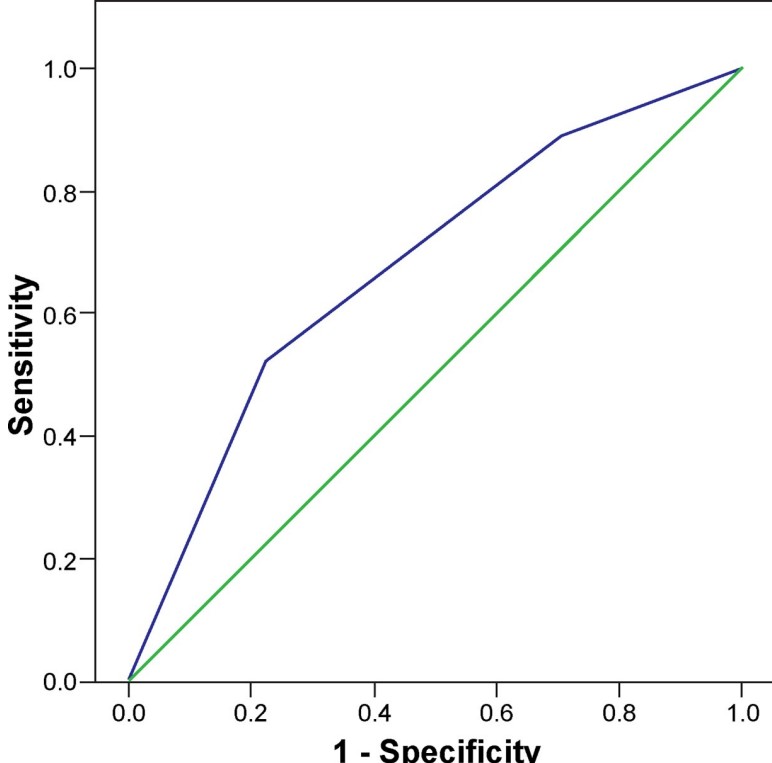

**Fig 4. Receiver operating characteristic curve of the plasma-leak score for predicting plasma leakage among dengue patients.**

affected by dengue [28]. The median time of the first visit to the hospital was 4 days of fever onset, similar to previous studies [8,16,26]. Differences in age and comorbid diseases were not observed between the two groups. However, previous studies have reported that diabetes, hypertension, cardiac disorders, and asthma patients were at increased risk for severe manifestations of dengue [9–12,29]. In our study, most clinical and laboratory findings were significantly different between plasma leakage patients and those without plasma leakage starting on

**Table 8. Prognostic values of the plasma-leak score for predicting plasma leakage.**

| Total score | With plasma leakage (n = 178) | Without plasma leakage (n = 121) | Sensitivity (95% CI) | Specificity (95% CI) | PPV (95% CI) | NPV (95% CI) | LR+ (95% CI) | LR- (95% CI) |
|---|---|---|---|---|---|---|---|---|
| ≥1 | 158 | 85 | 88.8 (83.2–93.0) | 29.8 (21.8–38.7) | 65.0 (62.1–67.9) | 64.3 (52.3–74.7) | 1.3 (1.1–1.4) | 0.4 (0.2–0.6) |
| ≥2 | 93 | 27 | 52.2 (44.6–59.8) | 77.7 (69.2–84.8) | 77.5 (70.6–83.2) | 52.5 (48.0–57.0) | 2.3 (1.6–3.4) | 0.6 (0.5–0.7) |
| 3 | 28 | 8 | 15.7 (10.7–21.9) | 93.4 (87.4–97.1) | 77.8 (62.3–88.1) | 43.0 (41.0–44.9) | 2.4 (1.1–5.0) | 0.9 (0.8–1.0) |

CI, confidence interval; LR+, likelihood ratio positive; LR-, likelihood ratio negative; NPV, negative predictive value; PPV, positive predictive value.

days 3 to 4 of fever onset. Bleeding was the most common condition associated with plasma leakage observed on days 5 to 8 of fever onset in this study.

Plasma leakage and bleeding are the hallmark of dengue hemorrhagic fever (DHF) and are associated with death from dengue [5,7]. This might be due to the linkage on the disease's pathogenesis via cytokine storm and antibodies response on the vascular endothelial cells and the hemostatic abnormalities after DENV infection [30,31]. Our study also showed that MAP and fluid balance were significantly higher among patients with plasma leakage. This might be because patients with plasma leakage received more fluid replacement than those without plasma leakage [5]. Currently, plasma leakage is the main pathophysiological hallmark of DHF [28]. According to the modified WHO/SEARO 2011 criteria, plasma leakage is the major criteria for distinguishing DHF from dengue fever (DF), without the necessity of bleeding [5,6]. Early recognition of plasma leakage can lead to appropriate fluid administration and then prevent the development of DSS, which is the most common cause of death from dengue [3–7]. However, most previous studies regarding plasma leakage predictors have varied results due to differences in study design, participant composition, and case definition of severe manifestations of dengue [8–19].

This study identified three independent factors associated with plasma leakage. They were BMI $\geq 25.0$ kg/m$^2$, PLT count $<100,000$/mm$^3$ on days 3 to 4 of fever onset, and AST or ALT $\geq 100$ U/l on days 3 to 4 of fever onset. As per systematic review and meta-analysis, most studies on the association of BMI and dengue severity were conducted in children. They showed that obese children had a higher risk for developing DHF or severe dengue compared to non-obese children; this is due to the stronger immune response in obese children than undernourished or normal weighted children [32,33]. A retrospective study from Malaysia showed that adult dengue patients with BMI $\geq 27.5$ kg/m$^2$ were at risk for elevated ALT, creatinine level, raised HCT, the occurrence of chills and rigors, high body temperature, and high systolic blood pressure [34].

The possible pathophysiological mechanisms for developing plasma leakage in obese patients with dengue might be due to endothelial dysfunction caused by the chronic release of pro-inflammatory cytokines from elevated leptin levels and production of reactive oxygen species (ROS). These would precipitate endothelial damage in addition to cytokine storm after DENV infection. Moreover, downregulation of AMP-protein kinase in obese patients could lead to lipid accumulation in the endoplasmic reticulum, facilitating viral replication [31,34,35].

We found significantly higher serum ALB levels among patients with plasma leakage than those without plasma leakage on days 1 to 2 of fever onset. This might be due to the well-nourished status of plasma leakage patients, which might build up a stronger immune response. However, serum ALB levels of patients with plasma leakage decreased on days 3 to 8 of fever onset and was significantly lower than those without plasma leakage. These might be due to the leakage of serum ALB into the extravascular compartment from a cytokine-mediated increase in vascular permeability by endothelial glycocalyx damage, which is the primary pathogenesis of plasma leakage [5,30,31]. In 2021, a systematic review and meta-analysis identifying risk predictors of progression to severe disease, defined as severe dengue or DHF during the febrile phase of dengue, was published. The authors showed that all included studies in the analysis consistently reported that patients who progressed to DHF had lower serum ALB levels than those who did not progress to DHF [36].

On days 3 to 4 of fever onset, PLT count $<100,000$/mm$^3$ was a predictor for plasma leakage similar to a multicenter retrospective study in Thailand [15]. Plasma leakage and thrombocytopenia has a link with the pathogenesis of the disease via cytokine storm and cross-reactive immunoglobulin M type of antibodies after DENV infection, which is the potential mechanism of vascular pathology and PLT destruction [31,37,38].

In addition, AST or ALT ≥100 U/l on days 3 to 4 of fever onset was also a predictor for plasma leakage. Similarly, previous reports showed that elevated transaminases were an independent factor associated with severe manifestations of dengue [12,13]. Hepatocytes and Kupffer cells in the liver are important targets of DENV, which results in direct damage of liver cells by apoptosis and release of pro-inflammatory cytokines, which also results in endothelial damage [39,40]. In our study, the dosage of acetaminophen recommended by the National Thai guidelines, which relies on the US Food and Drug Administration suggestion for the reduction of fever, was used. The maximum daily dose of acetaminophen for an adult is 3000 mg with a recommended dosage of 500 mg every 6 hours. In 2021, a systematic review and meta-analysis identifying risk predictors of progression to severe disease, defined as severe dengue or DHF during the febrile phase of dengue, showed that all included studies in the analysis consistently reported that higher levels of AST or ALT were associated with progression to severe disease [36]. Thus, it is suggested that the elevation of AST or ALT in our study accounted for the association with DENV infection rather than with acetaminophen-induced hepatoxicity.

Till date, few studies developed a predictive score for dengue severity in adults, including the dengue score for predicting pleural effusion and/or ascites [13] and the clinical risk score for prediction of severe dengue [41]. In our study, a plasma-leak score was developed for identifying plasma leakage using a score as 1 for each parameter, including BMI ≥25.0 kg/m$^2$, PLT count <100,000/mm$^3$ on fever days 3 to 4, and AST or ALT ≥100 U/l on fever days 3 to 4. The plasma-leak score had a good discriminative ability with AUROC of 0.677. The sensitivity for the occurrence of plasma leakage was 88.8% for a score ≥1. The specificity for the occurrence of plasma leakage rose to 77.7% score ≥2, and as high as 93.4% for 3. The PPV was also increased to 77.5% for a score ≥2 and 77.8% for 3. These predictors are simple routine parameters for the early identification of patients who are at risk for plasma leakage, and the plasma-leak score could help with risk stratification of dengue. The risk stratification could help physicians to provide close observation as well as early and appropriate management, to prevent the progression to DSS. Ultimately, these measures would help in reducing the hospital cost, cost to the patients, and healthcare personnel workload.

However, this study's limitations were that the study was conducted in a single-center, which is the referral center for tropical diseases in Bangkok, Thailand, and the plasma-leak score's external validity needs to be evaluated.

## Conclusions

Dengue patients with BMI ≥25.0 kg/m$^2$ or who presented with PLT count <100,000/mm$^3$, or AST or ALT ≥100 U/L on days 3 to 4 of fever onset are at risk for the occurrence of plasma leakage. Patients with a plasma-leak score ≥1 had high sensitivity (88.8%) for the development of plasma leakage, and those with a plasma-leak score of 3 had high specificity (93.4%) for plasma leakage. This simple and easily accessible clinical score might help physicians provide close observation with early and appropriate clinical management of dengue even in resource-limited settings.

## Acknowledgments

The authors thank all patients who participated in this study, the staff, doctors in charge, and nurses in the outpatients and inpatients department of the Hospital for Tropical Diseases in Bangkok. We also thank Ms. Akanitt Jittmittraphap (Department of Microbiology and Immunology, Faculty of Tropical Medicine, Mahidol University, Assistant professor) and Ms. Boongong Noochan (Clinical Infectious Diseases Research Unit, Department of Clinical Tropical Medicine, Faculty of Tropical Medicine, Mahidol University) for their help with this study.

We thank Dr. Pratap Singhasivanon (former Dean of the Faculty of Tropical Medicine, Mahidol University; Associate Professor) and Dr. Porntip Petchmitr (former Deputy Dean of the Faculty of Tropical Medicine, Mahidol University; Associate Professor) for their support of this study.

## Author Contributions

**Conceptualization:** Sutopa Talukdar, Vipa Thanachartwet, Varunee Desakorn, Supat Chamnanchanunt, Duangjai Sahassananda, Mukda Vangveeravong, Siripen Kalayanarooj, Anan Wattanathum.

**Data curation:** Sutopa Talukdar, Vipa Thanachartwet, Varunee Desakorn, Duangjai Sahassananda.

**Formal analysis:** Sutopa Talukdar, Vipa Thanachartwet, Varunee Desakorn, Duangjai Sahassananda, Mukda Vangveeravong, Siripen Kalayanarooj, Anan Wattanathum.

**Funding acquisition:** Vipa Thanachartwet, Varunee Desakorn.

**Investigation:** Sutopa Talukdar, Vipa Thanachartwet, Varunee Desakorn.

**Methodology:** Sutopa Talukdar, Vipa Thanachartwet, Varunee Desakorn, Supat Chamnanchanunt, Duangjai Sahassananda, Anan Wattanathum.

**Project administration:** Sutopa Talukdar, Vipa Thanachartwet, Varunee Desakorn.

**Resources:** Vipa Thanachartwet, Varunee Desakorn, Duangjai Sahassananda.

**Software:** Vipa Thanachartwet, Varunee Desakorn, Duangjai Sahassananda.

**Supervision:** Vipa Thanachartwet, Varunee Desakorn, Supat Chamnanchanunt, Duangjai Sahassananda, Mukda Vangveeravong, Siripen Kalayanarooj, Anan Wattanathum.

**Validation:** Sutopa Talukdar, Vipa Thanachartwet, Varunee Desakorn.

**Visualization:** Sutopa Talukdar, Vipa Thanachartwet, Varunee Desakorn, Supat Chamnanchanunt, Duangjai Sahassananda, Mukda Vangveeravong, Siripen Kalayanarooj, Anan Wattanathum.

**Writing – original draft:** Sutopa Talukdar, Vipa Thanachartwet, Varunee Desakorn, Supat Chamnanchanunt, Duangjai Sahassananda, Mukda Vangveeravong, Siripen Kalayanarooj, Anan Wattanathum.

**Writing – review & editing:** Sutopa Talukdar, Vipa Thanachartwet, Varunee Desakorn, Supat Chamnanchanunt, Duangjai Sahassananda, Mukda Vangveeravong, Siripen Kalayanarooj, Anan Wattanathum.

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
