## [Decision Letter · Decision Letter 0]

25 Jun 2021

PONE-D-21-01273

Predictors of plasma leakage among dengue patients in Thailand: a plasma-leak score analysis

PLOS ONE

Dear Dr. Thanachartwet,

Thank you for submitting your manuscript to PLOS ONE. After careful consideration, we feel that it has merit but does not fully meet PLOS ONE’s publication criteria as it currently stands. Therefore, we invite you to submit a revised version of the manuscript that addresses the points raised during the review process.

We look forward to receiving your revised manuscript.

Kind regards,

Sherief Ghozy, M.D.

Academic Editor

PLOS ONE

Journal Requirements:

Reviewers' comments:

Reviewer's Responses to Questions

**Comments to the Author**

1. Is the manuscript technically sound, and do the data support the conclusions?

Reviewer #1: Partly

2. Has the statistical analysis been performed appropriately and rigorously? 

Reviewer #1: Yes

3. Have the authors made all data underlying the findings in their manuscript fully available?

Reviewer #1: Yes

4. Is the manuscript presented in an intelligible fashion and written in standard English?

Reviewer #1: Yes

5. Review Comments to the Author

Reviewer #1: The authors conducted a prospective hospital-based study of confirmed adult dengue subjects to evaluate the predictors and the predictive score of plasma leakage.

The authors justified that plasma leakage in dengue is a significant contributor to disease severity, morbidity and mortality. Nearly half (47.7%) of their adult population, which is quite sizeable, developed plasma leakage. The independent factors associated with plasma leakage include BMI > 25 kg/m2, platelet count <100,000/mm3 on day 3 to 4 and liver enzymes > 100 U/l on day 3-4 of fever. Patients with plasma leak score of 1 had high sensitivity and a score of 3 had high specificity.

In general, the authors have done a thorough statistical analysis and managed to identify these 3 independent factors and the odds ratio of having a score of 1 or 2 or 3.

My comments:

1. There are some issues with the definition of plasma leakage; increase of HCT > 20%, clinical fluid accumulation and/or hypoalbuminemia (<3.5 g/L or decrease of 0.5 g from baseline). Hypoalbuminemia on its own is not sufficient evidence of plasma leakage. (Reference 1)

2. Targeting a single pathophysiology in a highly complex pathophysiological scenario as in severe dengue is quite tricky. Furthermore, only 4.4% of those with plasma leakage developed dengue shock. It is now quite well established among dengue experts (References 2-3) that dengue disease manifest as a spectrum of continuum of severity rather than as 2 disease entities which is suggested by the authors; DF and DHF which was also the premise of the 1997 WHO dengue case classification. In fact, their data also suggested that there could be a spectrum of disease severity. Although almost all (95.6%) of those with PL were hospitalized a significant number of those without PL (71.3%) were also hospitalized; so the question is – why were they hospitalized? What were the concerns doctors had to decide for admission? If, according to their line of thinking that PL is severe, then not having PL should not be hospitalized. Are there other factors that determine severity of disease that are not captured in this scoring system?

3. Did the authors account for the use of paracetamol in their evaluation of raised ALT/AST?

4. Although a score would be helpful for clinicians to recognise those at risk, having a cut-off value does not really help in the actual clinical picture. The concern is that clinicians would rely solely on the score rather than evaluate the patient in a clinical way, in the same way as having a cut-off for platelet count of <100,000 be considered for admission, as was in 1997 Dengue Case Classification.

5. The authors should suggest how to operationalise their findings in the clinical setting of a highly dynamic disease of diverse clinical manifestations.

References:

1. Tomashek KM, Wills B, See Lum LC, Thomas L, Durbin A, Leo YS, et al. Development of standard clinical endpoints for use in dengue interventional trials. PLoS Negl Trop Dis. 2018 Oct 4;12(10):e0006497.

2. Libraty DH, Young PR, Pickering D, Endy TP, Kalayanarooj S, Green S, et al. High circulating levels of the dengue virus nonstructural protein NS1 early in dengue illness correlate with the development of dengue hemorrhagic fever. J Infect Dis. 2002 Oct 15;186(8):1165-8. doi: 10.1086/343813.

3. Phuong LTD, Hanh TTT, Nam VS, Climate. Variability and Dengue Hemorrhagic Fever in Ba Tri District, Ben Tre Province, Vietnam during 2004-2014. AIMS Public Health. 2016 Sep 26;3(4):769-780. doi:10.3934/publichealth.2016.4.769.

Minor comments:

1. Table 2 has too many variables, some are not relevant. It would be good if the significant p values in tables 2 and 3 be highlighted.

2. Figures are another way of displaying the results, again it should be indicated with asterisk in the figures at which point the differences became significant.

6. PLOS authors have the option to publish the peer review history of their article (what does this mean?). If published, this will include your full peer review and any attached files.

Reviewer #1: **Yes: **Lucy Chai See Lum

---

## [Author Response · Author response to Decision Letter 0]

12 Jul 2021

Journal Requirements

Ans: Thank you very much for the suggestions. The manuscript was checked and corrected per PLOS ONE’s style requirements. The following changes are some of the corrected ones.

1. Figures: 

1.1. “Figure 1” was corrected to “Fig 1” as shown in the results (Page 11, Line 186).

1.2. “Figure 2a” was corrected to “Fig 2a” as shown in the results (Page 18, Line 236).

1.3. “Figure 2b” was corrected to “Fig 2b” as shown in the results (Page 24, Line 265).

1.4. “Figure 2c” was corrected to “Fig 2c” as shown in the results (Page 24, Line 268).

1.5. “Figure 3a and Figure 3b” was corrected to “Fig 3a and Fig 3b” as shown in the results (Page 24, Line 271). 

1.6. “Figure 3c” was corrected to “Fig 3c” as shown in the results (Page 24, Line 274 and 277). 

1.7. “Figure 4” was corrected to “Fig 4” as shown in the results (Page 32, Line 339). 

2. References: All references were carefully checked and corrected as journal requirement instructions.

Major comments

1. There are some issues with the definition of plasma leakage; increase of HCT > 20%, clinical fluid accumulation and/or hypoalbuminemia (<3.5 g/L or decrease of 0.5 g from baseline). Hypoalbuminemia on its own is not sufficient evidence of plasma leakage. (Reference 1)

Ans: Thank you very much for the comments. Plasma leakage syndrome or capillary leakage syndrome was first described by Clarkson in 1960 and is characterized by hypoalbuminemia, hemoconcentration, and hypovolemic shock [1]. Several mechanisms of plasma leakage have been proposed for several conditions. However, increased capillary permeability allows protein and fluid component of blood pass through the endothelial barrier and into the interstitium; this is the mechanism of plasma leakage in many medical conditions including systemic inflammatory response syndrome or sepsis, acute pancreatitis, anaphylaxis, snake bites, anthrax, brucellosis, and dengue. Currently, glycocalyx degradation caused by reactive oxygen species, enzymes, and various cytokines such as TNF-alpha and IFN-gamma, which allow plasma to leak out into the tissues, are proposed as mechanisms of plasma leakage in dengue [2,3]. According to the revised WHO 2011 dengue guidelines, plasma leakage is manifested by a rising HCT ≥20% above the baseline, a drop in hematocrit ≥20% from the baseline in convalescence, or evidence of plasma leakage such as pleural effusion, ascites or hypoproteinemia/hypoalbuminemia [2].

Currently, plasma leakage is the hallmark feature for distinguishing dengue fever (DF) from dengue hemorrhagic fever (DHF) including DHF grades I-II or non-shock DHF and DHF grades III-IV or dengue shock syndrome (DSS) [2,4]. Regarding the WHO 2009 dengue guidelines, severe dengue, defined as severe plasma leakage, leads to DSS and/or fluid accumulation with respiratory distress, severe bleeding, or severe organ involvement [5]. A previous study from Malaysia showed that the most common cause of death in dengue was DSS (73%) followed by severe organ involvement (69%) and severe bleeding (30%) [6]. 

The WHO statement on the global strategy for dengue prevention and control, 2012-2020, stated that “Mortality from dengue can be reduced to almost zero by implementing timely, appropriate clinical management, which involves early clinical and laboratory diagnosis, intravenous rehydration, staff training and hospital reorganization” [7]. Early clinical and laboratory diagnosis of plasma leakage is important for providing close observation and fluid management to prevent disease progress to the development of DSS, multi-organ failure, and death. For the above reasons, plasma leakage was then considered as an outcome of interest in our study. Thus, the rational of our study was to determine the predictors of plasma leakage and develop a predictive score for plasma leakage among dengue patients aged ≥15 years.

According to literature review, there are a number of studies on dengue among adults and children using the definition of plasma leakage similar to our study as shown in Table 1. 

 

Table 1. Lists of literature review for definition of plasma leakage which similar to our study

Clinical Studies Definition of plasma leakage

Lee IK, Liu JW, Chen YH, et al. Development of a simple clinical risk score for early prediction of severe dengue in adult patients. PLoS ONE. 2016; 11(5): e0154772. doi:10.1371/journal.pone.0154772 -Adults

-Retrospective study during 2002-2015 in Taiwan

-Plasma leakage defined as presence of hemoconcentration >20%, pleural effusion, ascites, and/or hypoalbuminemia

Khurram M, Qayyum W, Umar M, et al. Ultrasonographic pattern of plasma leak in dengue haemorrhagic fever. J Pak Med Assoc. 2016;66(3):260-4. -Adults

- Retrospective study during July- Dec 2013 in Pakistan

-Plasma leakage defined as an increase in Hct>20% above average for age or decrease in Hct>20% of baseline following fluid replacement therapy, pleural effusion, ascites, or hypoproteinemia

Nainggolan L, Wiguna C, Hasan I, Dewiasty E.Gallbladder wall thickening for early detection of plasma leakage in dengue infected adult Patients.

Acta Med Indones. 2018;50(3):193-199. -Adults

-Prospective study during 2011- 2012 in Indonesia

- Plasma leakage defined as ≥20% elevation of HCT from baseline or decrease in convalescence, evidence of pleural effusion, ascites or hypoproteinemia/ hypoalbuminemia

Thomas L, Broustea Y, Najioullah F, et al. Predictors of severe manifestations in a cohort of adult dengue patients. Journal of Clinical Virology. 2010;48: 96–99. -Adults

-Prospective study during 2005-2008 in France

- Plasma leakage defined as pleural effusion, ascites, hypoproteinemia and/or hemoconcentration

Thein TL, Leo YS, Lee VJ, et al. Validation of probability equation and decision tree in predicting subsequent dengue hemorrhagic fever in adult dengue inpatients in Singapore. Am J Trop MedHyg.2011; 85(5): 942–945.

doi:10.4269/ajtmh.2011.11-0149 -Adults

-Prospective study during 2004-2007 in Singapore

-Plasma leakage defined as hypoproteinemia, 20% change in HCT, or pleural effusion or ascites

Pongpan S, Wisitwong A, Tawichasri C et al. Development of dengue infection severity score. ISRN Pediatrics. 2013; doi: 10.1155/2013/845876 -Children

-Retrospective study during 2007-2010 in Thailand

-Plasma leakage defined as an increasein HCT ≥20% from previous HCT or signs of plasma leakage, such as pleural effusion or ascites, or evidence of hypoalbuminemia

Tantracheewathorn T, Tantracheewathorn S.

Risk factors of dengue shock syndrome in children. Med Assoc Thai 2007; 90(2): 272-7. -Children

-Case control study during 2003-2005 in Thailand

-Plasma leakage defined as a rise of HCT >20% from baseline, a drop in HCT >20% from baseline after volume replacement, pleural effusion, ascites, or hypoproteinemia

A previous study by Tomashek et al. in 2018, a reference suggested by the reviewer, aimed to develop standardized endpoints in dengue using a Delphi methodology-based query. This study worked towards the consensus opinion of 26 clinical trial dengue experts on those endpoints. Of 26 invitations that were sent out, there were 22 active respondents (defined as persons who answered at least one question) in the first round with 91% completing all non-conditional questions, 19 active respondents (94% completion rate) in the second round, and 18 active respondents (92% completion rate) in the third round. Professional sector activities included 27% industry/vaccine developers, 54% academia, 27% public health, 35% clinical sector, 35% governmental sector, and 4% non-governmental organization. Participants reported working in 27% North America, 12% South-East Asia, 8% South America, 8% Central America, 8% Western Pacific, 8% Europe, and 4% Africa. 

After the third round of inquiry, ≥70% agreement was reached on moderate and severe plasma leakage. One of the operational considerations for moderate plasma leakage revealed that hypoalbuminemia or hypoproteinemia was not sufficient evidence of plasma leakage [8]. However, the US Agency for Health Care Policy and Research (AHCPR) has recommended that the Delphi techniques provided the lowest level of evidence for making causal inferences, and are thus subordinate to meta-analysis, intervention study, and correlation studies. Furthermore, the experts on Delphi study can draw on various sources of information to make their judgments, and significant weaknesses also exist in the quality of the reporting [9]. 

In a recent systematic review and meta-analysis in 2021, to identify the risk predictors of progression to severe disease, defined as severe dengue or DHF during the febrile phase of dengue, Sangkaew et al. showed that all included studies in the analysis consistently reported that patients who progressed to DHF had lower serum albumin levels than those who did not progress to DHF. They reported low to moderate heterogeneity and no publication bias [10]. A previous retrospective study done among 5,332 patients with dengue admitted between 1995 and 1999 at the Children’s Hospital in Bangkok, Thailand, showed that patients with DHF and DSS had mean serum albumin of 4.5 and 4.3 g/dl, respectively before the occurrence plasma leakage, then decreased to 4.1 and 3.6 g/dl, respectively after plasma leakage [11]. Therefore, hypoproteinemia defined as serum albumin ≤3.5 g/dL or a decrease ≥0.5 g/dL below baseline was used as one of the criteria, or evidence, for determining plasma leakage in our study [4]; as reported in the study design and population subsection (Page 7, lines 120–121). 

In addition, this information was added in the discussion (Page 37, lines 413–418) as follows: “In 2021, a systematic review and meta-analysis identifying risk predictors of progression to severe disease, defined as severe dengue or DHF during the febrile phase of dengue, was published. The authors showed that all included studies in the analysis consistently reported that patients who progressed to DHF had lower serum ALB levels than those who did not progress to DHF [36].” 

References

1. Clarkson B, Thompson D, Horwith M, Luckey EH. Cyclical edema and shock due to increased capillary permeability. Am J Med. 1960;29:193-216. doi: 10.1016/0002-9343(60)90018-8. pmid: 13693909.

2. World Health Organization (WHO). Comprehensive guidelines for prevention and control of dengue and dengue hemorrhagic fever. Revised and expanded edition. New Delhi, India: WHO; 2011. Available from: http://www.searo.who.int/entity/vector_borne_tropical_diseases/documents/SEAROTPS60/en/. [Accessed 07 Aug 2017]

3. Lam PK, McBride A, Le DHT, Huynh TT, Vink H, Wills B, Yacoub S. Visual and biochemical evidence of glycocalyx disruption in human dengue infection, and association with plasma leakage severity. Front Med (Lausanne). 2020;7:545813. doi: 10.3389/fmed.2020.545813, pmid: 33178710

4. Kalayanarooj S. Clinical manifestations and management of dengue/DHF/DSS. Trop Med Health. 2011;39(4 Suppl): 83–87. doi:10.2149/tmh.2011-S10, pmid: 22500140

5. World Health Organization (WHO). Dengue guidelines for diagnosis, treatment, prevention and control. Geneva: WHO; 2009. Available from: https://www.who.int/tdr/publications/documents/dengue-diagnosis.pdf. [Accessed 26 Jun 2021]

6. Woon YL, Hor CP, Hussin N, Zakaria A, Goh PP, Cheah WK. A Two-Year review on epidemiology and clinical characteristics of dengue deaths in Malaysia, 2013-2014. PLoS Negl Trop Dis 2016;10(5): e0004575. https://doi.org/10.1371/journal.pntd.0004575, pmid: 27203726

7. World Health Organization (WHO). Global strategy for dengue prevention and control 2012-2020. Geneva: WHO; 2012. Available from: http://www.who.int/denguecontrol/9789241504034/en/. [Accessed 07 Aug 2017]

8. Tomashek KM, Wills B, See Lum LC, Thomas L, Durbin A, Leo Y-S, et al. Development of standard clinical endpoints for use in dengue interventional trials. PLoS Negl Trop Dis 2018;12(10): e0006497. https://doi.org/10.1371/journal.pntd.0006497, pmid: 30286085

9. Niederberger M, Spranger J. Delphi technique in health sciences: A map. Front Public Health. 2020;8:457. doi: 10.3389/fpubh.2020.00457, pmid: 33072683

10. Sangkaew S, Ming D, Boonyasiri A, Honeyford K, Kalayanarooj S, Yacoub S, et al. Risk predictors of progression to severe disease during the febrile phase of dengue: a systematic review and meta-analysis. Lancet Infect Dis. 2021:S1473-3099(20)30601-0. doi: 10.1016/S1473-3099(20)30601-0, pmid: 33640077

11. Kalayanarooj S, Chansiriwongs V, Nimmannitya S. Dengue patients at the Children's Hospital, Bangkok: 1995-1999 Review. New Delhi: WHO; 2002. Available from: https://apps.who.int/iris/handle/10665/163764. [Accessed 30 JUN 2021]

2. Targeting a single pathophysiology in a highly complex pathophysiological scenario as in severe dengue is quite tricky. Furthermore, only 4.4% of those with plasma leakage developed dengue shock. It is now quite well established among dengue experts (References 2-3) that dengue disease manifest as a spectrum of continuum of severity rather than as 2 disease entities which is suggested by the authors; DF and DHF which was also the premise of the 1997 WHO dengue case classification. In fact, their data also suggested that there could be a spectrum of disease severity. Although almost all (95.6%) of those with PL were hospitalized a significant number of those without PL (71.3%) were also hospitalized; so the question is – why were they hospitalized? What were the concerns doctors had to decide for admission? If, according to their line of thinking that PL is severe, then not having PL should not be hospitalized. Are there other factors that determine severity of disease that are not captured in this scoring system?

Ans: Thank you very much for the comments. The underlying pathophysiology for increased disease severity in dengue is explained by antibody-dependent enhancement (ADE). ADE is linked to the presence of non-neutralizing or sub-neutralizing levels of dengue virus (DENV)–reactive IgG induced by a primary infection, or acquired passively in newborns. Non-neutralizing or sub-neutralizing antibodies bind heterologous DENV to facilitate virus entry through Fc receptors expressed on target cells, such as monocytes, macrophages, and dendritic cells. A large infected cell mass results in elevated concentrations of acute-phase response proteins, cytokines, and chemokines; generation of immune complexes; and consumption of complement and release of split products. The host immunologic response is thought to create a physiological environment in tissues that promotes capillary permeability when the viral burden is in rapid decline. Thus, ADE results in a greater burden of infection, which induces imbalanced pro-inflammatory and anti-inflammatory responses, which are thought to induce capillary endothelial pathology and then plasma leakage, potentially leading to DSS [1,2]. A recent study demonstrated the evidence of visual and biochemical glycocalyx degradation in patients with DENV infection, with worse visual glycocalyx damage and higher plasma degradation products associated with more severe plasma leakage [3].

A previous study from Malaysia showed that the most common cause of death in dengue was DSS (73%) followed by severe organ involvement (69%) and severe bleeding (30%) [4]. Thus, plasma leakage is the most serious complication in dengue and is the hallmark feature for distinguishing DF from DHF including DHF grades I-II or non-shock DHF and DHF grades III-IV or DSS [5,6]. Regarding the WHO 2009 dengue guidelines, severe dengue was defined as severe plasma leakage leading to DSS and/or fluid accumulation with respiratory distress, severe bleeding, or severe organ involvement [7]. 

In our study (Table 2 and 3), most clinical and laboratory findings were significantly different between plasma leakage patients and those without plasma leakage starting on days 3 to 4 of fever onset. However, bleeding was the most common condition associated with plasma leakage observed on days 5 to 8 of fever onset in this study. Plasma leakage and thrombocytopenia has a link to the pathogenesis of the disease via cytokine storm and cross-reactive immunoglobulin M type of antibodies after DENV infection, which is the potential mechanism of vascular pathology and platelet destruction [8-10]. Currently, no specific pathway has been identified linking immunopathogenic events with microvascular permeability or thromboregulatory mechanisms [11,12]. Both the virus itself and dengue nonstructural protein 1 (NS1) are known to adhere to heparan sulfate, a key structural element of the glycocalyx, and increased urinary heparan sulfate excretion has been detected in children with severe infection [13,14]. Heparan sulfate may also function as an anticoagulant and contribute to the coagulopathy [2]. Loss of essential coagulation proteins probably plays a major role in the development of the typical coagulopathy, which is usually manifested as an increase in the partial thromboplastin time accompanied by low fibrinogen levels, but with little evidence of procoagulant activation [2]. However, the exact pathophysiology for increased disease severity in dengue are still unclear [2].

The WHO statement on the global strategy for dengue prevention and control, 2012-2020, stated that “Mortality from dengue can be reduced to almost zero by implementing timely, appropriate clinical management, which involves early clinical and laboratory diagnosis, intravenous rehydration, staff training and hospital reorganization” [15]. Early clinical and laboratory diagnosis of plasma leakage is important for providing close observation and fluid management in order to prevent the disease progress to DSS and then multi-organ failure and death, while severe bleeding can occur after patients with DSS develop organ failure during defervescence [4,5]. Therefore, plasma leakage was used as the outcome in our study [5,6]. 

Our study was conducted as a prospective observational study at the Hospital for Tropical Diseases, Faculty of Tropical Medicine, Mahidol University in Bangkok, Thailand, between March 2018 and February 2020, in order to determine predictors of plasma leakage and develop a predictive score for plasma leakage among dengue patients aged ≥15 years.

Literature review for predictors of plasma leakage among dengue patients aged ≥15 years reports several different parameters as shown in the introduction (Pages 4-5, Lines 59-77) including demographic characteristics of older age [8-11], gender [8,11], ethnicity [11], diabetes mellitus [9,10-12], hypertension [11], delayed hospitalization [9], secondary infection [9], clinical parameters of bleeding [8], abdominal pain [8,10], lethargy [8,9], or cough [8], and laboratory findings of HCT rising[12-14], thrombocytopenia [13,15], abnormal coagulation profile [14], raised liver enzymes [12,13], low serum albumin level [13,15], or thickening of the gall bladder wall [9]. Recent studies have added several new parameters, including procalcitonin [16], lactate [16,17], chymase [18], and cytokines [19], as plasma leakage predictors among dengue patients aged ≥15 years.

However, even though several parameters as stated above have been assessed in studies to be predictive of plasma leakage, some of these laboratory parameters may not be accessible in remote and resource-limited settings, where patients at risk for plasma leakage need to be identified, using simple clinical assessment methods and easily accessible laboratory investigations to improve healthcare utilization efficiency and save patients from unnecessary expenditure, loss of productivity, morbidity, and mortality associated with dengue.

All patients in our study were managed by their attending physicians according to the standard guidelines for dengue management [5,6], as shown in the “study design and population” subsection (Page 7, Lines 111-112). 

Our study was conducted as a prospective observational study, with the aim to determine predictors of plasma leakage and develop a predictive score for plasma leakage among dengue patients aged ≥15 years. Thus, clinical and laboratory parameters according to the standard guidelines for dengue management were used and the patients were managed by their attending physicians.

Therefore, this information were added in the introduction (Page 4, lines 49-52) as follows: “A previous study showed dengue shock syndrome (DSS) as the most common cause of death in adults and children with dengue accounted for 73%, followed by severe organ involvent (69%) and severe bleeding (30%) [3].” and in the study design and population (Page 7, lines 112–117) as follows: “Tests were performed to obtain data on routine monitoring parameters, including the day of fever, clinical condition, vital signs, and complete blood count, during the follow up of the patients. Other tests, for data on additional laboratory parameters including liver enzymes, serum ALB, and chest radiography, were performed according to the attending physicians’ instructions, as per the clinical condition of the patients. Urine output was recorded for patients treated at the IPD.”

Regarding our routine clinical practice, the National Thai guidelines for the management of adults with dengue are used for consideration of admission. Admission is considered if the dengue patient meets one of the following criteria [16]: 

1. Patient who cannot eat or drink, having persistent vomiting, abdominal pain, fatigue, lethargy, or fainting

2. Patient with abnormal bleeding

3. Patient with hypotension, narrow pulse pressure, or having diagnosis of DSS

4. Patient with a rising of HCT ≥20% above baseline, or HCT >45% in female, or >50% in male

5. Patient with platelet counts <50,000 /mm3 with abnormal bleeding, or having platelet counts <20,000 /mm3

6. Patient with AST or ALT ≥500 U/l

7. Patient with acute kidney injury, alter mental status, or cardiac arrhythmia 

8. Patients with pregnancy, elderly, BMI ≥35 kg/m2, underlying medical illness such as diabetes mellitus, hypertension, heart diseases, liver diseases, hematologic diseases and kidney diseases, or patients who receive anticoagulants/antiplatelets (these indications rely on attending physician’s consideration)

9. Patient who is unable to undergo follow-up at the outpatient department.

Therefore, several clinical conditions as mentioned above were considered for admission among patients with dengue in our clinical practice, not only the occurrence of plasma leakage.

References

1. Wilder-Smith A, Ooi EE, Horstick O, Wills B. Dengue. Lancet. 2019;393(10169):350-363. doi: 10.1016/S0140-6736(18)32560-1, pmid: 30696575.

2. Simmons CP, Farrar JJ, Nguyen vV, Wills B. Dengue. N Engl J Med. 2012;366(15):1423-32. doi: 10.1056/NEJMra1110265, pmid: 22494122.

3. Lam PK, McBride A, Le DHT, Huynh TT, Vink H, Wills B, Yacoub S. Visual and biochemical evidence of glycocalyx disruption in human dengue infection, and association with plasma leakage severity. Front Med (Lausanne). 2020;7:545813. doi: 10.3389/fmed.2020.545813, pmid: 33178710.

4. Woon YL, Hor CP, Hussin N, Zakaria A, Goh PP, Cheah WK. A Two-Year Review on Epidemiology and Clinical Characteristics of Dengue Deaths in Malaysia, 2013-2014. PLoSNegl Trop Dis 2016;10(5): e0004575. https://doi.org/10.1371/journal.pntd.0004575, pmid: 27203726

5. World Health Organization (WHO). Comprehensive guidelines for prevention and control of dengue and dengue hemorrhagic fever. Revised and expanded edition. New Delhi, India: WHO; 2011. Available from: http://www.searo.who.int/entity/vector_borne_tropical_diseases/documents/SEAROTPS60/en/. [Accessed 07 Aug 2017]

6. Kalayanarooj S. Clinical manifestations and management of dengue/DHF/DSS. Trop Med Health. 2011;39(4Suppl): 83–87. doi:10.2149/tmh.2011-S10, pmid: 22500140

7. World Health Organization (WHO). Dengue guidelines for diagnosis, treatment, prevention and control. Geneva: WHO; 2009. Available from: https://www.who.int/tdr/publications/documents/dengue-diagnosis.pdf. [Accessed 26 Jun 2021]

8. St John AL, Abraham SN, Gubler DJ. Barriers to preclinical investigations of anti-dengue immunity and dengue pathogenesis. Nat Rev Microbiol. 2013;11(6): 420-426. doi:10.1038/nrmicro3030, pmid: 23652323 

9. Lin CF, Lei HY, Liu CC, Liu HS, Yeh TM, Wang ST, et al. Generation of IgM anti-platelet autoantibody in dengue patients. J Med Virol. 2001;63(2): 143-149. doi:10.1002/1096-9071(20000201)63:2<143::AID-JMV1009>3.0.CO;2-L, pmid: 11170051

10. Soundravally R, Sankar P, Bobby Z, Hoti SL. Oxidative stress in severe dengue viral infection: association of thrombocytopenia with lipid peroxidation. Platelets. 2008;19(6): 447-454. doi:10.1080/09537100802155284, pmid: 18925513 

11. Michel CC, Curry FE. Microvascular permeability. Physiol Rev. 1999;79(3):703-61. doi: 10.1152/physrev.1999.79.3.703, pmid: 10390517.

12. Levick JR, Michel CC. Microvascular fluid exchange and the revised Starling principle. Cardiovasc Res. 2010;87(2):198-210. doi: 10.1093/cvr/cvq062,pmid: 20200043

13. Chen Y, Maguire T, Hileman RE, Fromm JR, Esko JD, Linhardt RJ, Marks RM. Dengue virus infectivity depends on envelope protein binding to target cell heparan sulfate. Nat Med. 1997;3(8):866-71. doi: 10.1038/nm0897-866, pmid: 9256277

14. Avirutnan P, Zhang L, Punyadee N, Manuyakorn A, Puttikhunt C, Kasinrerk W, et al. Secreted NS1 of dengue virus attaches to the surface of cells via interactions with heparan sulfate and chondroitin sulfate E. PLoS Pathog. 2007;3(11):e183. doi: 10.1371/journal.ppat.0030183, pmid: 18052531

15. World Health Organization (WHO). Global strategy for dengue prevention and control 2012-2020. Geneva: WHO; 2012. Available from: http://www.who.int/denguecontrol/9789241504034/en/. [Accessed 07 Aug 2017]

16. Department of Medical Services, Ministry of Public Health in Thailand. Diagnosis and management of adults with dengue (in Thai). Nonthaburi: Ministry of Public Health in Thailand; 2020. Available from: https://www.dms.go.th/backend//Content/Content_FIle/Bandner_(Small)/Attach/25640302103903AM_CPG%20Adult%20Dengue.pdf. [Accessed 01 Jul 2021] 

3. Did the authors account for the use of paracetamol in their evaluation of raised ALT/AST?

Ans: Thank you very much for the comments. In our study, the dosage of acetaminophen recommended by the National Thai guidelines relies on the US Food and Drug Administration suggestion for the reduction of fever. The maximum daily dose of acetaminophen for an adult is 3000 mg with a recommended dosage of 500 mg every 6 hours. A recent systematic review and meta-analysis identifying the risk predictors of progression to severe disease, defined as severe dengue or DHF during the febrile phase of dengue, showed that all included studies in the analysis consistently reported that higher levels of AST or ALT were associated with progression to severe disease [1]. Since the patients in our study were given acetaminophen as recommended by the National Thai guidelines, which relies on the US Food and Drug Administration suggestion, the elevation of AST/ALT in our study would account for association with dengue viral infection rather than with acetaminophen-induced hepatoxicity.

Therefore, this information was added in the discussion (Page 38, lines 429–439) as follows: “In our study, the dosage of acetaminophen recommended by the National Thai guidelines, which relies on the US Food and Drug Administration suggestion for the reduction of fever, was used. The maximum daily dose of acetaminophen for an adult is 3000 mg with a recommended dosage of 500 mg every 6 hours. In 2021, a systematic review and meta-analysis identifying risk predictors of progression to severe disease, defined as severe dengue or DHF during the febrile phase of dengue, showed that all included studies in the analysis consistently reported that higher levels of AST or ALT were associated with progression to severe disease [36]. Thus, it is suggested that the elevation of AST/ALT in our study accounted for the association with dengue viral infection rather than with acetaminophen-induced hepatoxicity.” 

References

1. Sangkaew S, Ming D, Boonyasiri A, Honeyford K, Kalayanarooj S, Yacoub S, et al. Risk predictors of progression to severe disease during the febrile phase of dengue: a systematic review and meta-analysis. Lancet Infect Dis. 2021:S1473-3099(20)30601-0. doi: 10.1016/S1473-3099(20)30601-0, pmid: 33640077.

4. Although a score would be helpful for clinicians to recognize those at risk, having a cut-off value does not really help in the actual clinical picture. The concern is that clinicians would rely solely on the score rather than evaluate the patient in a clinical way, in the same way as having a cut-off for platelet count of <100,000 be considered for admission, as was in 1997 Dengue Case Classification.

Ans: Thank you very much for the comments. In dengue endemic areas, the WHO guidelines are used to guide the management of patients with dengue with slight modification according to the baseline health status of the respective community, experience of health care personnel, and hospital facilities in each country. Primary triage has to be performed by a trained health care personnel for assessing patients with dengue. For the first visit, the baseline parameters including age, gender, underlying medical illness, duration of fever, warning signs, weight, height, vital signs, and complete blood count were routinely performed to identify patients with severe disease. The follow-up parameters, including day of fever, clinical condition, urine output, vital signs, and complete blood count were routinely performed. The additional laboratory parameters including blood glucose, liver function test, renal function test, coagulogram, chest radiography, electrocardiogram, and blood for lactate, etc., were assessed during follow-up as required based on the patients’ condition, particularly patients with DSS. 

In our study, multivariate analysis showed that three independent factors associated with plasma leakage, including body mass index ≥25.0 kg/m2, platelet count <100,000 /mm3 on fever days 3 to 4, and aspartate aminotransferase or alanine aminotransferase ≥100 U/l on fever days 3 to 4, were predictors for plasma leakage. Patients with a plasma-leak score ≥1 had high sensitivity (88.8%), and those with a plasma-leak score of 3 had high specificity (93.4%) for plasma leakage occurrence. 

The parameters, including body mass index, platelet count and liver enzymes are simple, and easily accessible for predicting plasma leakage in clinical practice. Additionally, the plasma-leak score could be integrated into the routine clinical practice of health care personnel in dengue endemic areas. This is because the score can be used in any set up for risk stratification of dengue; thereby, close observation with early and timely appropriate clinical dengue management can be provided, to prevent progression to DSS.

However, this study's limitations were that the study was conducted in a single-center, which is the referral center for tropical diseases in Bangkok, Thailand, and the plasma-leak score’s external validity needs to be evaluated; as shown in the discussion (Page 39, Lines 455-457).

We believe that physicians would not rely only on the plasma-leak score for evaluation of patients with dengue because in routine clinical practice, the WHO guidelines are still used to guide the management of patients with dengue, with slight modification for appropriateness in each country. The plasma-leak score could be integrated into the routine clinical practice of health care professionals, and thereby, help with risk stratification, provision of close observation, with early and timely appropriate clinical dengue management, to prevent progression to DSS.

Dengue patients who presented with platelet count <100,000 /mm3 are at risk for the occurrence of plasma leakage in our study, which was not for the indication of admission. 

5. The authors should suggest how to operationalise their findings in the clinical setting of a highly dynamic disease of diverse clinical manifestations.

Ans: Thank you very much for the pertinent comments. The first visit of patients with dengue had a median (IQR) of 4.0 (3.0-5.0) days of fever onset, similar to previous studies, as shown in the discussion (Page 35, Lines 365-366) as follows: “The median time of the first visit to the hospital was 4 days of fever onset, similar to previous studies [8,16,26].” 

Our results showed that body mass index ≥25.0 kg/m2, platelet count <100,000 /mm3 on fever days 3 to 4, and aspartate aminotransferase or alanine aminotransferase ≥100 U/l on fever days 3 to 4 were predictors for plasma leakage. Patients with a plasma-leak score ≥1 had high sensitivity (88.8%) and those with a plasma-leak score of 3 had high specificity (93.4%) for plasma leakage occurrence. 

These predictors are simple routine parameters for the early identification of patients who are at risk for plasma leakage, and the plasma-leak score could help with risk stratification of patients with dengue. These could help the physicians with close observation and early and appropriate management, to prevent progression to DSS, which will ultimately reduce the hospital cost, cost to the patients, and healthcare personnel workload. 

However, this study's limitations were that the study was conducted in a single-center, which is the referral center for tropical diseases in Bangkok, Thailand, and the plasma-leak score’s external validity needs to be evaluated; as shown in the discussion (Page 39, Lines 455-457).

Therefore, the text “fever days 3 to 4” was added in the discussion (Pages 38, Lines 442-445) as follows: “In our study, a plasma-leak score was developed for identifying plasma leakage using a score as 1 for each parameter, including BMI ≥25.0 kg/m2, PLT count <100,000 /mm3 on fever days 3 to 4, and AST or ALT ≥100 U/l on fever days 3 to 4.” and the other information was added in the discussion (Page 39, lines 449–454) as follows: “These predictors are simple routine parameters for the early identification of patients who are at risk for plasma leakage, and the plasma-leak score could help with risk stratification of dengue. The risk stratification could help physicians to provide close observation as well as early and appropriate management, to prevent the progression to DSS. Ultimately, these measures would help in reducing the hospital cost, cost to the patients, and healthcare personnel workload.”

Minor comments

Question #1. Table 2 has too many variables, some are not relevant. It would be good if the significant p values in tables 2 and 3 be highlighted.

Ans: Thank you very much for the valuable comments and suggestions. Variables included in Table 2 are common presentation in adults with dengue. Respiratory tract symptoms (20-45%) and diarrhea (25-35%) could be observed in adults with dengue [1-5]. The tables included in the manuscript was checked, and they follow the journal’s requirement and instructions.

References

1. Tantawichien T. Dengue fever and dengue haemorrhagic fever in adolescents and adults. Paediatr Int Child Health. 2012 May;32(Suppl 1):22-7. doi: 10.1179/2046904712Z.00000000049, pmid: 22668446

2. Taylor WR, Fox A, Pham KT, Le HNM, Tran NTH, Tran GV, et al. Dengue in adults admitted to a referral hospital in Hanoi, Vietnam. Am J Trop Med Hyg. 2015;92(6):1141-1149. doi: 10.4269/ajtmh.14-0472, pmid: 25918201

3. Thomas L, Moravie V, Besnier F, Valentino R, Kaidomar S, Coquet LV, et al. Clinical presentation of dengue among patients admitted to the adult emergency department of a tertiary care hospital in Martinique: implications for triage, management, and reporting. Ann Emerg Med. 2012;59(1):42-50. doi: 10.1016/j.annemergmed.2011.08.010, pmid: 21903297.

4. Thanachartwet V, Oer-Areemitr N, Chamnanchanunt S, Sahassananda D, Jittmittraphap A, Suwannakudt P, et al. Identification of clinical factors associated with severe dengue among Thai adults: a prospective study. BMC Infect Dis. 2015;15:420. doi: 10.1186/s12879-015-1150-2, pmid: 26468084

5. Thanachartwet V, Desakorn V, Sahassananda D, Jittmittraphap A, Oer-Areemitr N, Osothsomboon S, et al. Serum Procalcitonin and Peripheral Venous Lactate for Predicting Dengue Shock and/or Organ Failure: A Prospective Observational Study. PLoS Negl Trop Dis. 2016;10(8):e0004961. doi: 10.1371/journal.pntd.0004961, pmid: 27564863

Question #2. Figures are another way of displaying the results, again it should be indicated with asterisk in the figures at which point the differences became significant.

Ans: Thank you very much for the important suggestions. The asterisks are shown in Figs 2 and 3 to show the significant differences, and they have been corrected as you suggested.

---

## [Editor Report · Decision Letter 1]

15 Jul 2021

Predictors of plasma leakage among dengue patients in Thailand: a plasma-leak score analysis

PONE-D-21-01273R1

Dear Dr. Thanachartwet,

We’re pleased to inform you that your manuscript has been judged scientifically suitable for publication and will be formally accepted for publication once it meets all outstanding technical requirements.

Kind regards,

Sherief Ghozy, M.D.

Academic Editor

PLOS ONE
---

## [Editor Report · Acceptance letter]

21 Jul 2021

PONE-D-21-01273R1 

Predictors of plasma leakage among dengue patients in Thailand: a plasma-leak score analysis 

Dear Dr. Thanachartwet:

I'm pleased to inform you that your manuscript has been deemed suitable for publication in PLOS ONE. Congratulations! Your manuscript is now with our production department. 

Kind regards, 

on behalf of

Dr. Sherief Ghozy 

Academic Editor

PLOS ONE